# Far-Future Climate Projection of the Adriatic Marine Heatwaves: a kilometre-scale experiment under extreme warming

Cléa Denamiel[1,2]

[1]Ruđer Bošković Institute, Division for Marine and Environmental Research, Bijenička cesta 54, 10000 Zagreb, Croatia

[2]Institute for Adriatic Crops and Karst Reclamation, Put Duilova 11, 21000 Split, Croatia

*Correspondence to*: Cléa Denamiel (cdenami@irb.hr)

**Abstract.** The impact of a far-future extreme warming scenario on Adriatic marine heatwave (MHW) characteristics—including intensity, duration, spatial extent, and associated environmental drivers—is assessed using the Adriatic Sea and Coast (AdriSC) kilometre-scale atmosphere-ocean model. The main aim of this study is to evaluate the added value and limitations of the Pseudo-Global Warming (PGW) approach, used to force the far-future AdriSC simulation, in projecting Adriatic MHWs. In line with existing knowledge, the results indicate a significant increase in MHW intensity along with a notable expansion in spatial coverage, particularly in the central and eastern Adriatic. Seasonal patterns show that the most intense MHWs occur between May and September, with events extending into late autumn and early winter under extreme warming. This study also reveals several novel insights. First, the Po River plume is identified as a key factor for the onset and decline of MHWs. Lower river discharges are associated with intense MHW onset, while higher discharges aid in heat dissipation during decline phases. As, air-sea heat fluxes are demonstrated to play a critical role in MHW onset along the plume, these findings suggest that MHWs are more likely to develop and persist under low Po River discharge conditions, when water clarity increases and solar radiation absorption is enhanced due to reduced suspended sediments and organic matter. Second, the study identifies a gap in MHW activity, potentially linked to the Eastern Mediterranean Transient, highlighting the influence of natural variability on MHW dynamics. However, no correlation is found with the Ionian-Adriatic Bimodal Oscillating System, suggesting the need for further research on oceanographic influences. Consequently, the PGW approach is found to effectively captures the thermodynamic changes influencing the MHWs in the Adriatic Sea despite potentially oversimplifying future MHW dynamics as it assumes stationarity in climate signals. Finally, these findings underscore the urgent need for adaptive strategies to mitigate the impacts of intensified MHWs on marine ecosystems and coastal communities, particularly in vulnerable nearshore areas. Future research should incorporate ensemble of high-resolution projections and assess additional climate stressors to provide a more comprehensive understanding of Adriatic MHWs under future warming.

## 1 Introduction

In the Mediterranean basin, the past decade has witnessed an alarming escalation in extreme heatwave events, both on land
and in the sea (Pastor et al., 2024). These heatwaves have soar beyond historical norms, with their unprecedented scale posing
significant challenges to ecosystems and human societies alike. In particular, Tejedor et al. (2024) have highlighted the
alarming potential of anthropogenic climate change to transform rare, extreme atmospheric heatwaves—once characterized by
10,000-year return periods, such as those observed in 2023 and 2024 in the Mediterranean—into regular occurrences happening
as frequently as every four years under extreme warming scenarios. This startling shift not only threatens to exceed the adaptive
capacity of human populations in the Mediterranean basin, as emphasized by Masselot et al. (2025) and Bujosa Mateu et al.
(2024), but also carries profound consequences for oceanic ecosystems. Such marine impacts include habitat degradation,
biodiversity loss, and shifts in ecological dynamics, as previously noted by Smale et al. (2019).

Recent studies underscore the importance of utilizing high-resolution datasets capable of resolving mesoscale and sub-
mesoscale processes, such as downwelling caused by eddies or waves, to comprehensively evaluate the impacts of marine
heatwaves (MHWs) from the surface to the subsurface layers (Dayan et al., 2022; Garrabou et al., 2022). These datasets are
essential not only for identifying the role of MHWs in triggering mortality events (Garrabou et al., 2019; Smith et al., 2024)
but also for addressing the specific needs of local stakeholders, such as aquaculture farms, which are highly sensitive to
localized oceanic temperature changes (Zemunik Selak et al., 2024). Despite this, a standardized best practice for selecting
models or reanalyses capable of accurately depicting MHW events at sub-regional and local scales remains elusive.

In this context, the relevance of the Pseudo-Global Warming (PGW) approach (Schär et al., 1996; Denamiel et al., 2020a)
becomes particularly evident. This method involves modifying existing reanalysis datasets by superimposing climatological
differences between historical and future regional or global climate simulations to drive high-resolution coastal models at
kilometre-scale resolutions. In the atmosphere, this approach has already proven effective in capturing the substantial increase
in heatwave frequency, duration, and intensity projected under future scenarios. Applications of the PGW methodology in this
context have provided valuable insights across Europe (e.g., Fischer and Schär, 2010), Asia (e.g., Imada et al., 2019), and
North America (e.g., Patricola and Wehner, 2018), where it accounts for localized feedback mechanisms such as soil moisture
depletion and urban heat island effects. However, a key limitation of the PGW approach lies in its assumption of stationarity
in the climate signal, which risks oversimplifying evolving climate dynamics. This limitation is particularly significant for
phenomena governed by non-linear climate change processes, as highlighted by Brogli et al. (2023).

The PGW approach has predominantly been applied to atmospheric studies and this gap in research underscores the necessity
to evaluate its effectiveness in modelling MHWs. Understanding whether this approach can accurately simulate the oceanic
conditions associated with MHWs is crucial, as it would enable researchers to assess the impacts of climate change on marine
ecosystems and coastal communities more effectively. Given the increasing frequency and severity of Mediterranean MHWs,
as highlighted by recent studies (Martín et al., 2024), applying the PGW methodology to marine environments could provide
valuable insights into future oceanic conditions under various climate scenarios. This advancement would enhance the research

community capacity to predict and mitigate the adverse effects of MHWs on marine biodiversity and human activities dependent on marine resources.

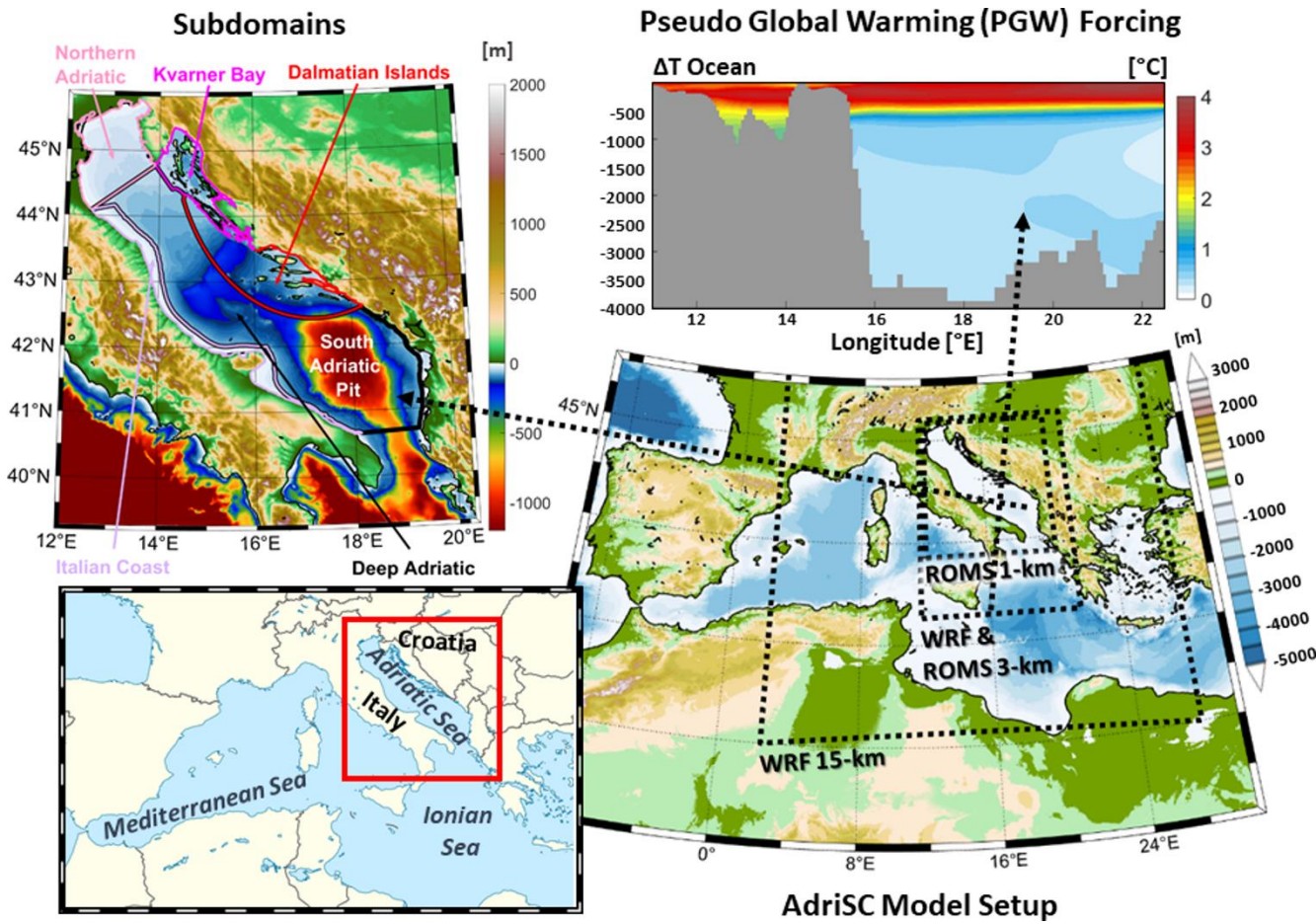

**Figure 1. Spatial coverage and horizontal resolution of the different grids used in the AdriSC climate model Setup including the topo-bathymetry of the AdriSC 1-km model with the locations of the 5 subdomains (coloured polygons) used in this study and the Pseudo-Global Warming temperature ocean forcing imposed in the AdriSC 3-km model southern boundary for the extreme warming simulation.**

In this study, the kilometre-scale Adriatic Sea and Coast (AdriSC) climate model (Denamiel et al., 2019; Figure 1) is thus used to understand how the PGW approach compares to traditional downscaling methods in terms of the characterization of the marine heatwaves within the Adriatic basin. The ability of the AdriSC model to simulate the present and future decadal atmosphere-ocean dynamics have already been assessed with many studies demonstrating the added value of such a kilometre-scale approach (Denamiel et al., 2020a, b, 2021a, b, 2022, 2025; Pranić et al., 2021, 2023, 2024; Tojčić et al., 2023, 2024). Consequently, the present study focuses on analysing and characterizing in detail the far-future impacts of an extreme warming scenario on the Adriatic MHW events. The article is structured as follows. Section 2 describes the AdriSC model and the

analytical methods used; Section 3 assesses and discusses the impacts of climate change on Adriatic MHW events from the basin to the subdomain scales; Section 4 evaluates the strengths and limitations of the PGW approach for characterizing far-future MHW; and Section 5 presents the main conclusions of the study.

## 2 Model and Methods

### 2.1 Adriatic Sea and Coast (AdriSC) Model

The Adriatic Sea and Coast (AdriSC) climate model (Denamiel et al., 2019) was designed to enhance the representation of atmospheric and oceanic dynamics within the Adriatic basin, offering improved modelling of the local processes compared to existing Mediterranean Regional Climate System Models (RCSMs). Built on the Coupled Ocean–Atmosphere–Wave–Sediment Transport (COAWST) framework (Warner et al., 2010), it couples the Weather Research and Forecasting (WRF; Skamarock et al., 2005) atmospheric model with the Regional Ocean Modeling System (ROMS; Shchepetkin and McWilliams,

2009). In the AdriSC climate simulations, as described and illustrated in Denamiel et al. (2025) and Figure 1, the WRF model employs two nested grids with 15-km and 3-km resolutions, while ROMS utilizes grids with 3-km and 1-km resolutions. The model uses vertical terrain-following coordinates, with 58 atmospheric levels refined near the surface (Laprise, 1992) and 35 oceanic levels optimized near the sea-surface and ocean floor (Shchepetkin and McWilliams, 2009).

    The AdriSC suite is implemented and rigorously validated on the high-performance computing infrastructure of the European

Centre for Medium-Range Weather Forecasts (ECMWF). Additional details about its configuration are available in Denamiel et al. (2019, 2021a) and Pranić et al. (2021).

    This research evaluates the effects of climate change on MHWs using two 31-year AdriSC climate simulations: a historical simulation spanning 1987–2017 and a far-future extreme warming scenario for 2070–2100, based on Representative Concentration Pathway (RCP) 8.5 (referred to as the RCP 8.5 simulation).

For the historical run, initial and boundary conditions are provided to the WRF 15-km model using 6-hourly ERA-Interim reanalysis data at 0.75° resolution (Dee et al., 2011) and to the ROMS 3-km model using MEDSEA reanalysis from the Mediterranean Forecasting System at 1/16° resolution (Simoncelli et al., 2019). Prior evaluations of the AdriSC historical run confirm its ability to accurately replicate the Adriatic's multi-decadal dynamics (Denamiel et al., 2022, 2025; Pranić et al., 2021, 2024).

### 2.2 Pseudo-Global Warming (PGW) methodology

    As described and illustrated in Denamiel et al. (2025) and Figure 1, for the RCP 8.5 simulation, the Pseudo-Global Warming (PGW) approach (Schär et al., 1996; Denamiel et al., 2020a) is used to adjust the historical forcing dataset by incorporating climatological changes from the LMDZ4-NEMOMED8 RCSM (Hourdin et al., 2006; Beuvier et al., 2010). Specifically, atmospheric variables such as air temperature, relative humidity, and wind components from ERA-Interim are modified using

differences between the 2070–2100 and 1987–2017 periods under RCP 8.5 ($\Delta T$, $\Delta RH$, $\Delta U$, $\Delta V$, respectively). These changes

generate 6-hourly three-dimensional atmospheric forcing for all 366 days of the year, which are then applied to the WRF 15-km model in the PGW simulation. Similarly, oceanic variables, including temperature, salinity, and currents, are adjusted using the climatological differences to produce daily three-dimensional oceanic forcing for ROMS 3-km ($\Delta T$ ocean, $\Delta S$ ocean, $\Delta U$ ocean, $\Delta V$ ocean, respectively). This process ensures that each simulated year inherits the synoptic conditions of the historical reanalysis while embedding the projected climatological shifts.

However, the pseudo-global warming (PGW) approach has several limitations, including simplified atmosphere-ocean dynamics, the assumption of stationarity, perturbations in initial and boundary conditions, resolution constraints, and neglected feedback mechanisms.

The PGW method used in the AdriSC climate model to predict marine heatwaves (MHWs) in the Adriatic Sea incorporates both thermodynamic changes—such as temperature, salinity, and humidity—and dynamic adjustments (i.e., wind and ocean currents). Consequently, the inaccuracies in representing circulation patterns critical for heatwave development, as highlighted by Xue et al. (2023), are somewhat mitigated. However, the PGW approach assumes that the relationship between large-scale climate drivers and regional patterns remains constant over time (i.e., the same climatological changes are applied every year; Brogli et al., 2023). This assumption can lead to potential misrepresentations of future MHW characteristics.

As Xue et al. (2023) point out, the effectiveness of the PGW method also depends on how perturbations (i.e., climatological changes) are applied to initial and boundary conditions. Inconsistent or inappropriate perturbations can result in significant variations in simulated outcomes, affecting the reliability of heatwave projections. Additionally, Heim et al. (2023) note that PGW is constrained by the resolution of the driving data and the capabilities of the regional model, which can impact the accurate representation of localized heatwave events. In this study, the initial and boundary conditions are derived from a coupled atmosphere-ocean Med-CORDEX regional climate model with a resolution of approximately 15 km, which is likely sufficient to capture the main dynamical properties of the Mediterranean Sea. However, the very short spin-up period (two months) for the AdriSC RCP 8.5 simulation is likely to influence the results, particularly in the first two to three years of the simulation.

Finally, by focusing on imposed large-scale changes, the PGW approach may overlook regional feedback processes, such as land-atmosphere interactions, which can influence heatwave intensity and frequency (Heim et al., 2023). A key limitation of the AdriSC climate model is that it employs a one-way coupling between the atmosphere and the ocean—i.e., the sea-surface temperature from the ocean model is not fed back into the atmospheric model.

**2.3 Methods**

The heatwaveR package designed by Schlegel et al. (2017) and based on the Marine Heatwave (MHW) definitions of Hobday et al. (2016, 2018) is used to extract mean and extremum intensities, duration, cumulative heat, onset and decline intensity rates and starting date of each MHW event. The MHW events are defined for AdriSC daily sea-surface temperatures above the daily 90[th] percentile (hereafter MHW threshold, illustrated in Figure 2) for at least 5 consecutive days. As recommended in Hobday et al. (2016), the thresholds are defined over the 31-year long historical and RCP 8.5 periods and the difference

between the daily threshold and mean climatology is used to calculate the different MHW categories. Additionally, in this study, the MHW events extracted with the heatwaveR package are only selected if they cover more than 5 % of the Adriatic basin. The area of the MHW events is calculated by considering that each model grid point covers a 1 km$^2$ surface.

It should also be noted that, for the RCP 8.5 scenario, the MHW are extracted with the thresholds extracted from the RCP 8.5 results to avoid extracting a nearly continuous MHW over the 31-year long period of the RCP 8.5 simulation (Tojčić et al., 2024). However, to simplify the comparison with the historical results, all the RCP 8.5 intensities presented in this article (except if mentioned explicitly) are shifted by $\Delta T_{c\lim}$, the difference between the RCP 8.5 and historical daily mean climatologies (hereafter RCP 8.5 MHW threshold, as illustrated in Figure 2). This methodological adjustment is widely adopted in marine heatwave (MHW) research to enable meaningful intercomparisons across different climatological baselines (e.g., Deser et al., 2024). The rationale behind this approach is to separate the influence of background warming from the intrinsic properties of MHWs. Without such an adjustment, the intensities under RCP 8.5 would reflect both the elevated baseline temperatures and the heatwave anomalies, making it difficult to isolate the climate change signal. By applying a shift equal to the difference between the RCP 8.5 and historical mean climatologies, the analysis focuses on the MHWs relative to their respective mean states, thereby allowing a clearer and more direct comparison of their characteristics across time periods. Additionally, although the RCP 8.5 MHW threshold exhibits pronounced seasonal variability, as shown in Figure 2, it remains effectively constant when considering annual distributions.

All annual and monthly probability density functions of the historical and RCP 8.5 intensities are normalized to have a unity area following a kernel-smoothing method (Bowman and Azzalini, 1997) evaluated for 100 equally spaced points.

The category of the MHW is calculated at each grid point of the ROMS 1-km grid using the maximum intensity reached during the MHW while the mean intensity at each model grid point is used to characterize the MHW events. For each event the Cumulative Heat Index (CHI) is calculated by integrating the cumulative heat (i.e., mean intensity multiply by duration) over the area of the MHW. Animations of the historical and RCP 8.5 MHW events (mean intensity, duration and category) extracted with heatwaveR over the 31-years of the simulations are presented as supplementary material (Movies S1 and S2, respectively). For each selected MHW event and only for the grid points corresponding to the area covered by the MHW, the daily ocean vertical temperatures for the 35 sigma layers and the daily minimum and maximum air temperatures (hereafter air temperatures during night and day) are extracted from the AdriSC historical and RCP 8.5 simulations. Additionally, the total air-sea net heat fluxes ($Q_{net}$) are calculated as described in Annexe A1 of Denamiel et al. (2025). Their contribution ($dSST_{Q_{net}}$, hereafter) to the change in sea-surface temperature anomaly ($dSST_A$, hereafter) during MHW onset and decline phases (between the first day and the peak of the MHW event and between the peak and the last day of the MHW event, respectively) is derived along the Adriatic coastline (between 5 and 50 m depth were most of the aquaculture is located) with the Schlegel et al. (2021) methodology previously used in the Mediterranean by Denaxa et al. (2024; see equations 1 to 3). Finally, the vertically integrated Ocean Heat Content (OHC), which corresponds to the energy absorbed by the ocean and stored as internal energy

for an indefinite time period (Hansen et al., 2011; von Schuckmann et al., 2016; Trenberth et al., 2018), is computed within the upper 20 m layers.

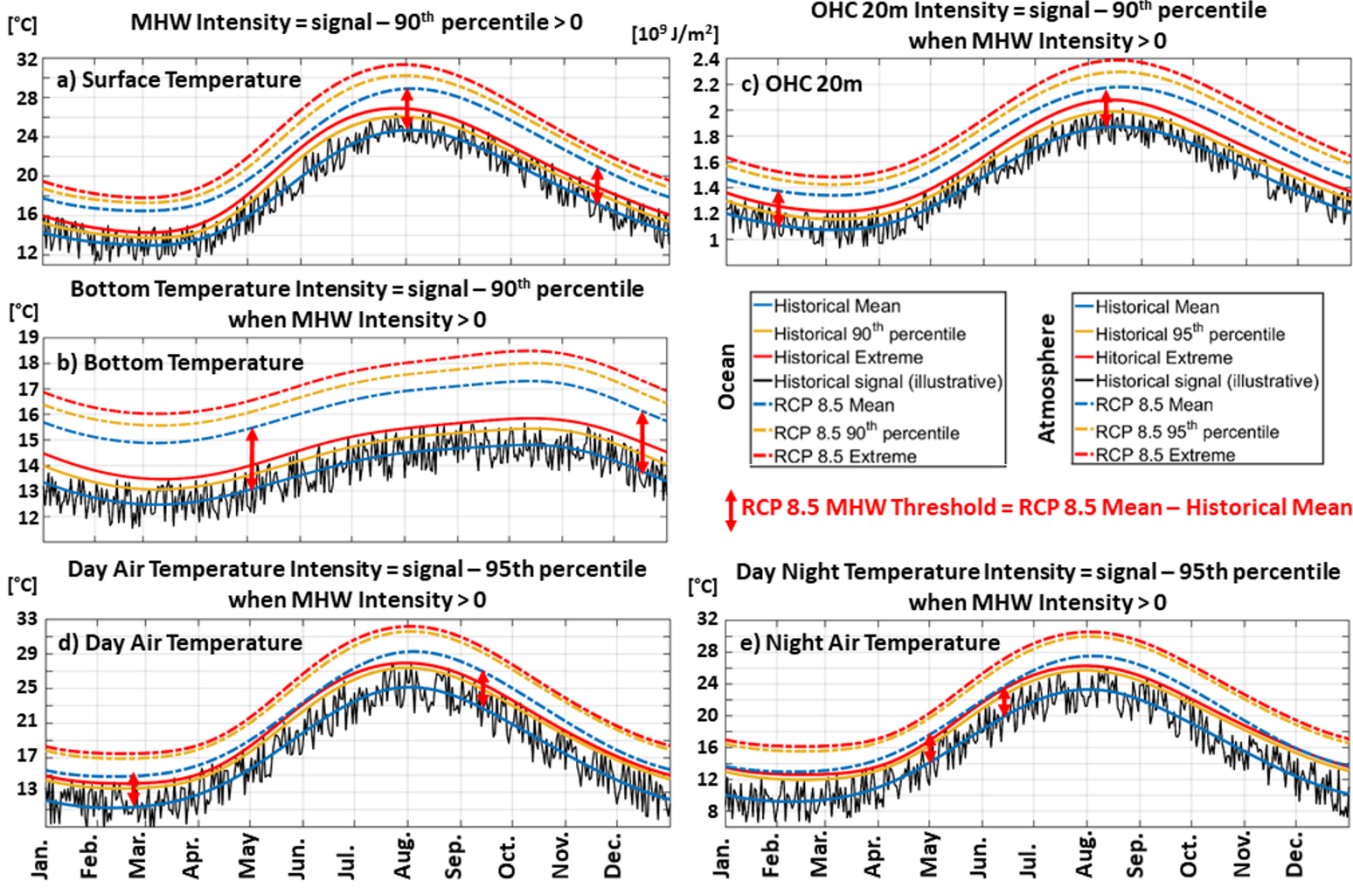

**Figure 2. Illustration of the methodology used to extract the daily intensities of the sea-surface temperature (a), sea-bottom temperature (b), Ocean Heat Content (OHC) at 20 m (c), day air temperature (d) and night air temperature (e) during the Marine Heat Waves (MHW) extracted from both the Historical and RCP 8.5 simulations.**

All the other intensities are calculated by removing the daily historical 90th percentile for the ocean bottom temperatures as well as the OHCs and the daily historical 95th percentile (i.e., threshold generally used to detect daily atmospheric heatwaves; European State of the Climate, 2023) for the air temperatures during night and day from these results and averaging them over the duration the MHW (illustrated in Figure 2). All thresholds are extracted from the 31-year long period of the historical and RCP 8.5 simulations.

A specific methodology has been applied for the evaluation of the influence of Adriatic Sea dynamics on MHW variability discussed in Section 4. Following Amaya et al. (2023) and Smith et al. (2025), both MHWs and the contributions of the total air-sea heat fluxes ($dSST_{Q_{net}}$) to the change in sea-surface temperature anomaly ($dSST_A$) have been recalculated using

detrended sea-surface temperature signals. In this study, trends are calculated using the Theil-Sen estimation method (Mondal et al., 2012), which is robust to outliers and often more accurate than simple linear regression when applied to skewed or heteroskedastic data. This method also performs competitively with non-robust least squares regression in terms of statistical power, even for normally distributed datasets. Additionally, the non-parametric Mann-Kendall test (originally proposed by Mann, 1945; further developed by Kendall, 1975; and Gilbert, 1987) is employed to detect the presence of monotonic (linear or non-linear) trends by evaluating whether a time series shows a consistent increase, decrease, or no change.

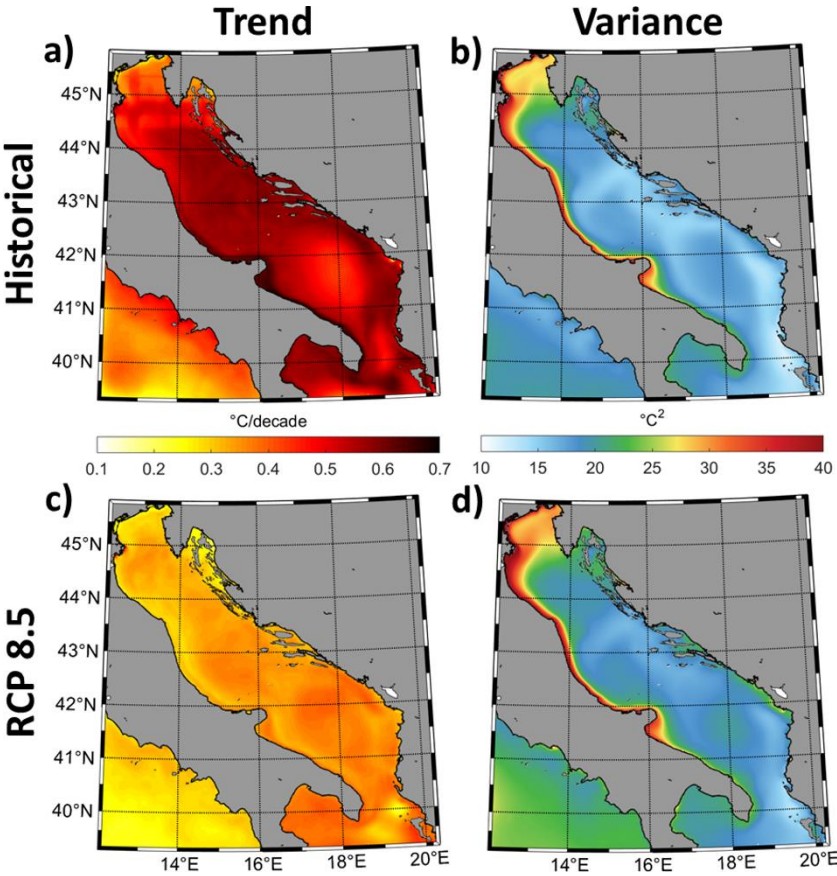

**Figure 3. Trend (a, c) and variance (b, d) for the sea-surface temperature over the Adriatic basin derived for the historical (a, b) and RCP 8.5 (c, d) 31-year long periods.**

For the AdriSC historical and RCP 8.5 simulations, Tojčić et al. (2023, 2024) conducted a comprehensive analysis of sea-surface temperature trends, variance, and extremes, showing that all identified trends are statistically significant. Notably, as illustrated in Figure 3, the rate of surface ocean warming is 40 % higher during the historical period than under the far-future RCP 8.5 scenario. Nonetheless, Tojčić et al. (2024) demonstrated that in the far-future period, the Adriatic Sea is projected to experience at least 20 additional days of extreme heat per month compared to historical conditions. This increase is primarily driven by surface temperature anomalies (i.e., differences between RCP 8.5 and historical conditions) exceeding 3 °C,

especially in coastal regions. Regarding sea-surface temperature variance (Fig. 3b, d), an average increase of 15 % is observed across the entire Adriatic Sea, with localized increases of up to 25 % along the south-eastern coastal regions. These changes further highlight the relevance of the methodology adopted to extract and analyse MHWs. For the comparison between historical and RCP 8.5 conditions, the consistent climatological baseline enables the accurate identification of MHWs, while the additive adjustment applied to the RCP 8.5 scenario helps isolate the heatwave signal from background warming. For the analysis of the influence of ocean dynamics on MHWs, the removal of the large sea-surface temperature trends is necessary to properly characterise the link between dynamical features and extreme events.

Finally, as highlighted in Juza et al. (2022) and Dayan et al. (2022, 2023), subdividing the Mediterranean in subdomains is crucial to capture the high spatial variability of ocean response to global warming and extreme events. In this study, the main advantage of using a 1-km resolution ocean model to extract MHWs is to be able to characterize the environmental conditions during these events for 5 different subdomains chosen for their particular dynamics. The chosen four nearshore subdomains (Fig. 1), are the Northern Adriatic (NA) and the Kvarner Bay (KB) where most of the Adriatic Dense water is formed, the Dalmatian Islands (DI) where a lot of tourism and fishery activities take place, and, finally, the Italian Coast (IC) along the path of the Po River plume with depths below 50 m. The chosen offshore subdomain covers all the remaining Adriatic Sea areas, with depths above 50 m, including both the Southern Adriatic and Jabuka Pits.

## 3 Results

### 3.1 Characterization of the Marine Heatwave events

MHWs are first characterised over the Adriatic Sea for the 31-year period of the historical and RCP 8.5 simulations through the spatial distributions of both the number of events (Fig 4a, b) and the mean accumulative intensity (Fig. 4c, d) as well as the temporal evolution of the Cumulative Heat Index (CHI; Fig 4e, f). Notably, the spatial distribution of the number of MHW events remains nearly identical under both historical and far-future extreme warming conditions (RCP 8.5 simulation). The Deep Adriatic subdomain (Fig. 1) experiences the highest number of events (35–50) while the nearshore areas, between the coast and 50 m depth, are affected by significantly fewer events (20–35).

In terms of mean cumulative intensity, the spatial patterns under historical and far-future extreme warming conditions are also similar, with the highest and lowest values located within the Dalmatian Islands and Northern Adriatic subdomains, respectively. However, a substantial increase in cumulative intensity is projected under far-future extreme warming, with values about six times higher (30–70 °C day) than under the historical conditions (5–12 °C day) across the entire Adriatic basin.

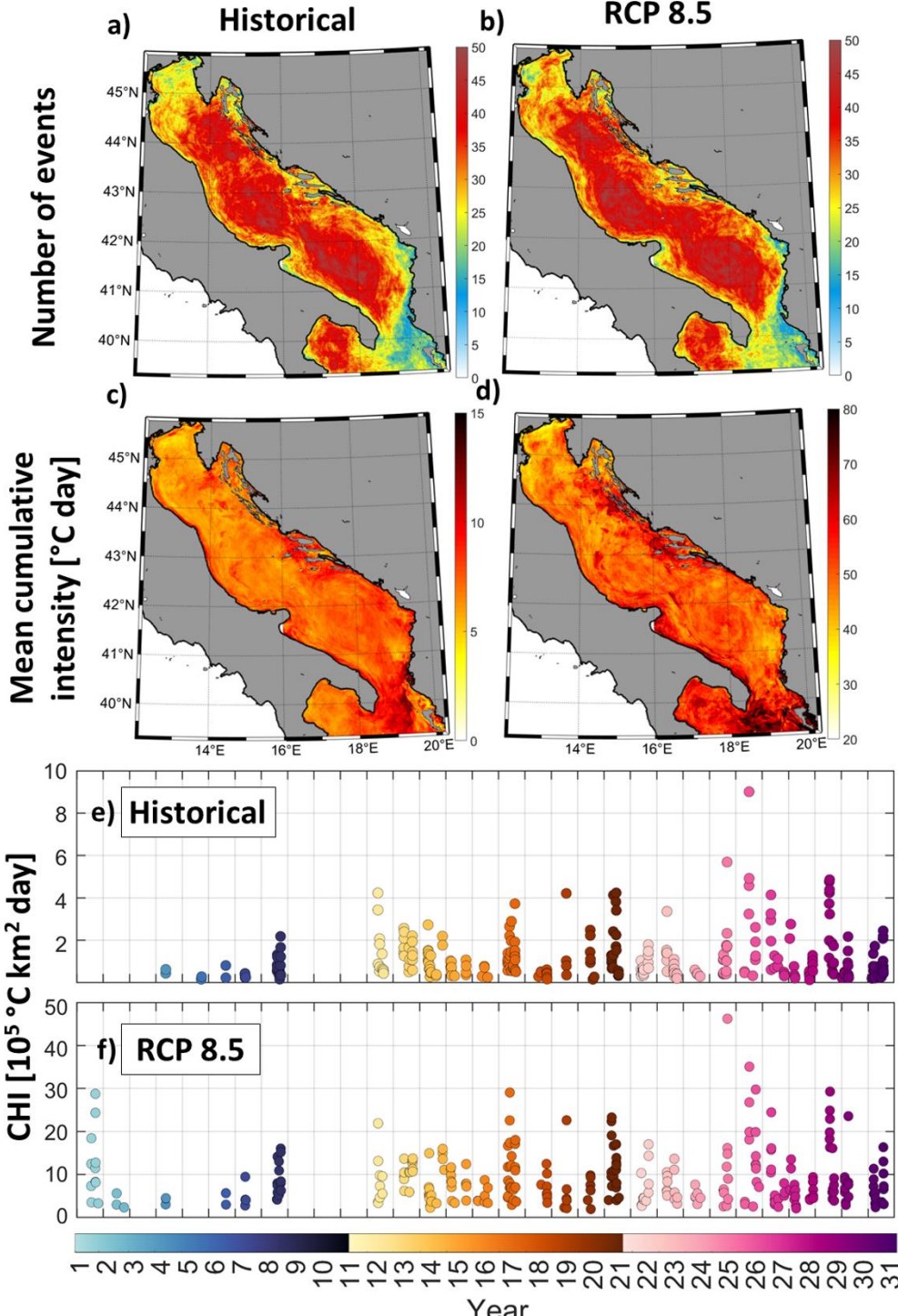

**Figure 4. Spatial distributions of the total number of marine heatwave (MHW) events (a, b) and their associated mean cumulative intensity (c, d) across the Adriatic Sea for the historical period (a, c) and the RCP 8.5 scenario (b, d). Panels e and f show the Time series of Cumulative Heat Intensity (CHI) over the 31-year historical and RCP 8.5 periods (e, f).**

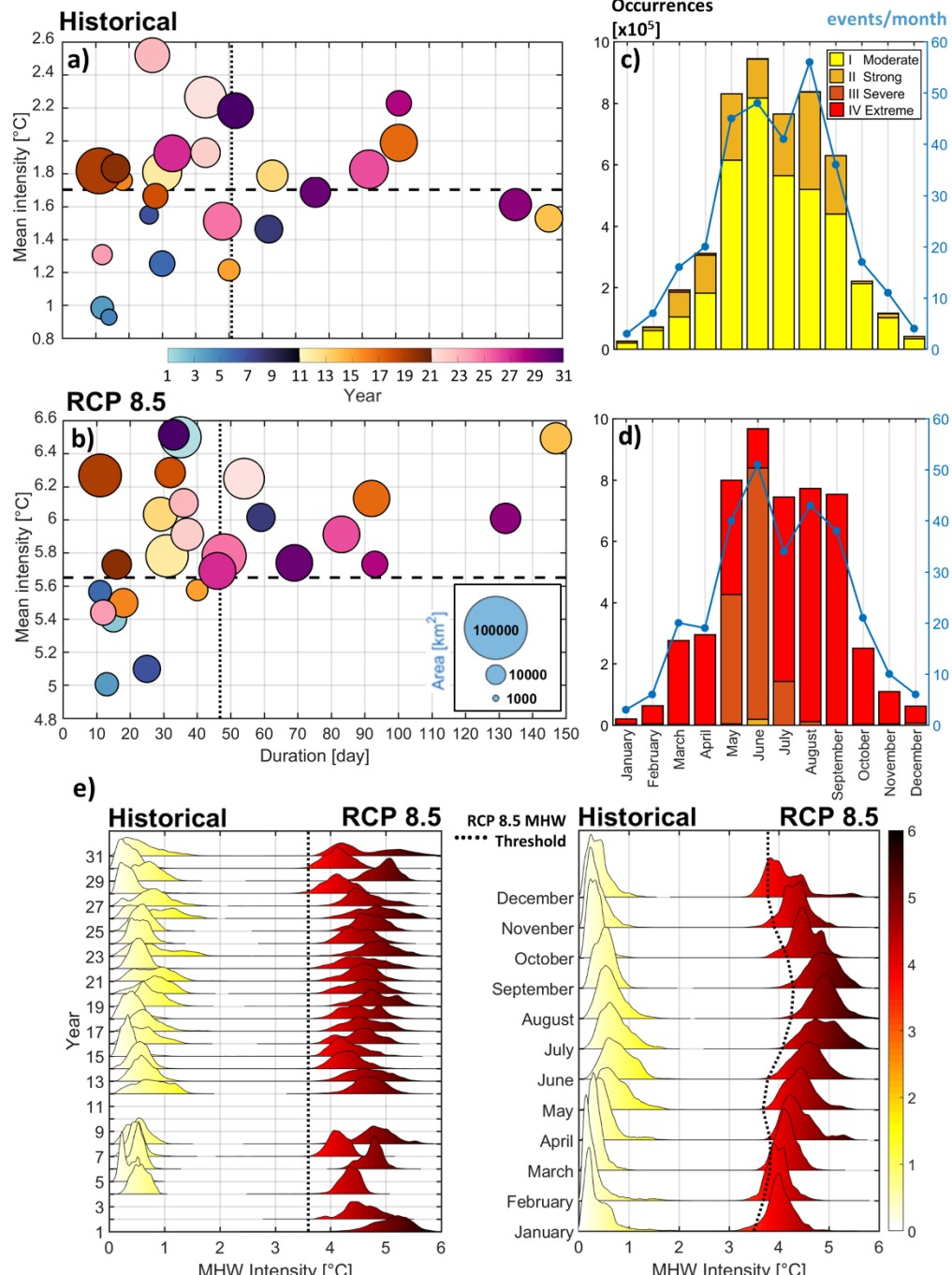

**Figure 5.** Scatter plots of the yearly maximum value of the MHW mean intensity versus the maximum yearly value of the duration of the MHW with the size of the circles depending on the maximum yearly value of the area covered by the MHW during the historical (a) and RCP 8.5 (b) 31-year long periods. Monthly distributions of both the MHW categories---with each occurrence representing one cell of the model---presented as histograms and the number of MHW events during the historical (c) and RCP 8.5 (d) 31-year long periods. Yearly (left) and monthly (right) probability density distributions of the MHW mean intensity during the

**historical and RCP 8.5 31-year long periods (e). All probability density distributions are normalized to unit area to facilitate direct comparison.**

Regarding the CHI, historical values remain relatively low in early years (below 2 x $10^5$ °C km$^2$ day for the first decade of the simulation). A gradual increase is observed over time with values exceeding 3.7 x $10^5$ °C km$^2$ day (the 95$^{th}$ percentile of the historical distribution) and reaching up to 9 x $10^5$ °C km$^2$ day in the final decade, indicating a rising trend in MHW severity. Under far-future extreme warming conditions, CHI values are nearly an order of magnitude higher than in the historical period, consistently exceeding 10 x $10^5$ °C km$^2$ day and peaking at 48 x $10^5$ °C km$^2$ day in the final decade. Interestingly, unlike in

the historical period, no clear increasing trend in CHI is modelled over the 31-year duration of the RCP 8.5 simulation. However, the strong MHWs modelled in the first year of the RCP 8.5 simulation may result from imbalances introduced by the PGW perturbations in the initial conditions. Notably, under both historical and RCP 8.5 conditions, no MHWs occur during years 9–11, while the most extreme values are reached during years 25 and 26. As explained in Denamiel et al. (2025), the large-scale atmospheric and oceanographic patterns that force the boundaries of the AdriSC WRF 15-km and ROMS 3-km

grids under historical conditions are also present in the RCP 8.5 simulation. Consequently, the 3-year gap in MHW occurrence observed under both historical and RCP 8.5 conditions is likely linked to an extraordinary atmospheric or oceanographic event, which will be further discussed in Section 4.

Further characterization of MHWs under historical and RCP 8.5 conditions is conducted using scatter plots displaying yearly maximum values of MHW mean intensity, duration, and area (Fig. 5a, b), monthly distributions of MHW categories and event

counts over the 31-year period (Fig. 5c, d), and yearly and monthly probability density distributions of MHW mean intensity (Fig. 5e). Under both historical and far-future extreme warming conditions, the yearly maximums of MHW duration are on average 40 and 46 days, respectively, the maximum affected areas mostly stay below 50,000 km², and the yearly maximums of mean intensity have an average of 1.7 °C and 5.6 °C, respectively. In both simulations, the longest-duration events (150 days) occur in year 14, while the smallest-area events (areas below 10,000 km$^2$) are recorded in the first 10 years. The highest

maximum intensities are reached in year 25, corresponding to 2011, in the historical period and in years 1, 14, and 31 under RCP 8.5.

The yearly density plots reveal that no MHWs occur in the first three years of the historical simulation. Initially, intensity distributions resemble Gaussian distributions, with means ranging from 0.36 to 0.57 °C during the first 10 years. Over time, they become increasingly right skewed toward the 2 °C mark, with means between 0.35 and 0.78 °C. In contrast, under far-

future extreme warming, intensity distributions are right skewed toward 6 °C from the start of the simulation, with means ranging between 4.13 and 5.1 °C over the 31-year period. As previously noted, no MHWs are detected in years 9–11 in either simulation.

Regarding monthly heatwave categories, MHWs shift from moderate and strong in the historical period to severe and extreme under extreme warming, though their seasonal distributions remain similar, with the most extreme events occurring between

May and September. The maximum number of MHW events per month (exceeding 50) occurs in August during the historical period, predominantly under strong conditions, whereas under RCP 8.5, it occurs in June, primarily under severe conditions.

The highest occurrences of extreme MHW conditions in the RCP 8.5 scenario are recorded between July and September, coinciding with the highest occurrences of strong conditions in the historical period.

The monthly distribution of MHW intensity reveals that, on average, the most intense heatwaves occur in June under historical conditions but shift to August under far-future extreme warming. Additionally, more extreme MHW intensities (with up to a 1.2 °C increase in temperature range) are expected to occur in November and December under RCP 8.5 compared to historical conditions.

Finally, the number of events per year as well as the distributions of the MHW mean intensity, duration and area over the 31 years of the simulations are used to compare the results obtained under historical and RCP 8.5 conditions (Fig. 6). Importantly, for consistency in comparison, the RCP 8.5 mean intensities are not adjusted for the difference between RCP 8.5 and historical temperature climatologies.

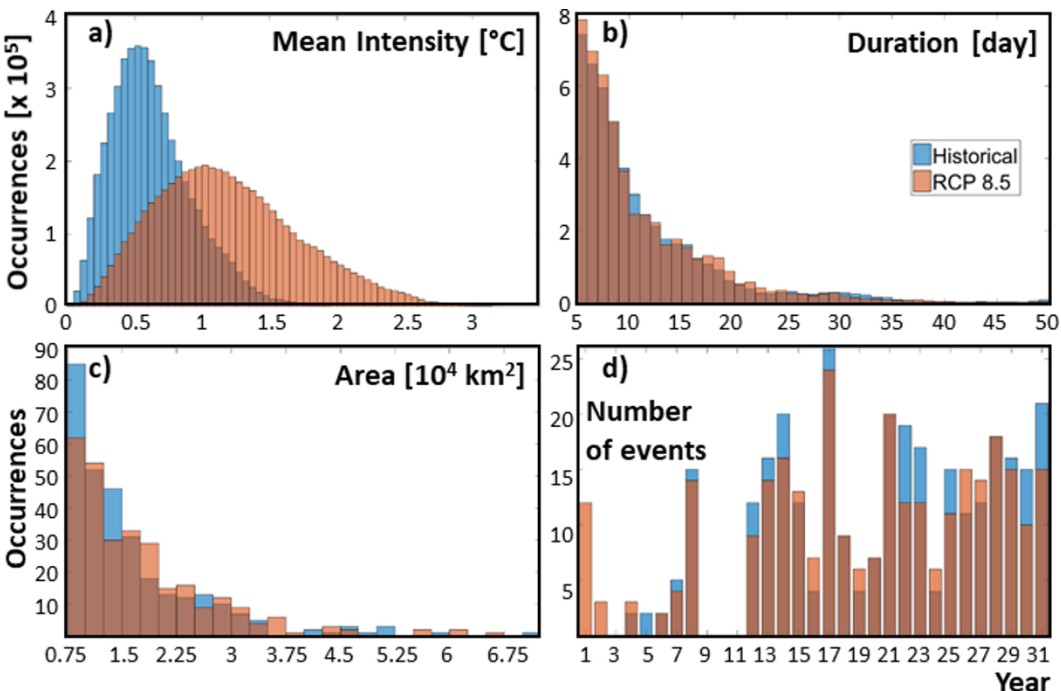

**Figure 6. Distributions of the MHW mean intensity as originally extracted with the heatwaveR package (a), duration (b) and area (c) as well as yearly time series of the number of MHW events defined over the 31 years for the historical and RCP 8.5 conditions.**

Notably, the AdriSC results indicate that, compare to historical conditions, MHWs under RCP 8.5 conditions are significantly more intense with median mean intensities of 0.59 and 1.14 °C, and 75th percentiles of 0.80 and 1.53 °C, respectively. They also have a larger spatial extent, with median areas of about 13,000 and 15,000 km$^2$, and 75th percentiles of about 19,000 and 21,000 km$^2$, respectively. However, their duration remains similar, with median values of 8 days and 75th percentiles of 13 days under both scenarios.

Interestingly, MHWs are slightly less frequent under RCP 8.5 conditions, with a total of 290 events compared to 304 events under historical conditions. This difference can be attributed to the fact that the historical trends are approximately 40 % higher than those under RCP 8.5 conditions (Fig. 3) and will be further discussed in Section 4. Furthermore, over the last 20 years of the simulations when MHWs occur annually, the correlations between median yearly variations in MHW intensity, area and duration under historical and RCP 8.5 conditions reach 0.92, 0.44 and 0.91, respectively. These high correlations between

historical and RCP 8.5 MHW intensity, area, and duration are likely a consequence of the PGW approach—which assumes that the relationship between large-scale climate drivers and regional weather patterns remains constant over time—rather than realistic temporal features.

## 3.2 Environmental conditions during the Marine Heatwave events

The environmental conditions during historical and RCP 8.5 marine heatwaves (MHWs) are analysed across five subdomains:

Northern Adriatic, Kvarner Bay, Italian Coast, Dalmatian Islands, and Deep Adriatic (Fig. 1). These conditions are presented as overall distributions and monthly climatologies of the median, 25[th], and 75[th] percentiles for MHW intensity, bottom temperature, and Ocean Heat Content (OHC) at 20 m depth in the ocean (Fig. 7), as well as for day and night air temperature intensities in the atmosphere (Fig. 8).

Regarding the distributions of MHW mean intensities under historical conditions, median and 75[th] percentile values are

approximately 0.60 °C and 0.80 °C for the Northern Adriatic (0.60 °C, 0.80 °C), Kvarner Bay (0.62 °C, 0.86 °C), Dalmatian Islands (0.58 °C, 0.80 °C), and Deep Adriatic (0.57 °C, 0.79 °C) subdomains. In contrast, the Italian Coast subdomain shows lower values, at 0.53 °C and 0.71 °C, respectively. Under the RCP 8.5 scenario, median and 75[th] percentile values increase significantly, ranging from 4.37 °C and 4.62 °C in the Northern Adriatic to 4.70 °C and 5.00 °C in the Deep Adriatic. The proportion of occurrences above heatwave thresholds are derived from the distributions of bottom temperature, OHC at 20 m,

day and night air temperature intensities (Table 1). Results show that, during historical surface MHW events, 28–32 % of bottom ocean temperatures and 45–54 % of OHC at 20 m exceed their heatwave thresholds in the Northern Adriatic, Kvarner Bay, Dalmatian Islands, and Deep Adriatic subdomains. However, in the Italian Coast subdomain, these proportions are significantly higher, with 42 % of bottom ocean temperatures and 62 % of OHC at 20 m exceeding their thresholds.

For RCP 8.5 surface MHW events, only 20–23 % of bottom temperatures exceed their thresholds in the Kvarner Bay and

Dalmatian Islands subdomains, while proportions remain similar to historical conditions for the other subdomains. In contrast, the proportion of OHC at 20 m exceeding its threshold increases under RCP 8.5 compared to historical conditions, particularly in Kvarner Bay (60 % vs. 54 %), Dalmatian Islands (65 % vs. 45 %), and the Deep Adriatic (51 % vs. 47 %). Additionally, across all subdomains, a higher percentage of night and day air temperatures exceed their thresholds under RCP 8.5 (37–44 % and 43–50 %, respectively) compared to historical conditions (28–37 % and 32–44 %, respectively). Notably, the presence of

a double peak in the distribution of ocean bottom temperature intensities in the Deep Adriatic subdomain (at about 0.5 °C and 3 °C; Fig. 7b) is linked to the fact that, under the RCP 8.5 scenario, the shallow areas of the Adriatic Sea are expected to warm by up to 3.5 °C, while the deepest parts only by up to 0.5 °C (Denamiel et al., 2025). Consequently, as the ocean bottom

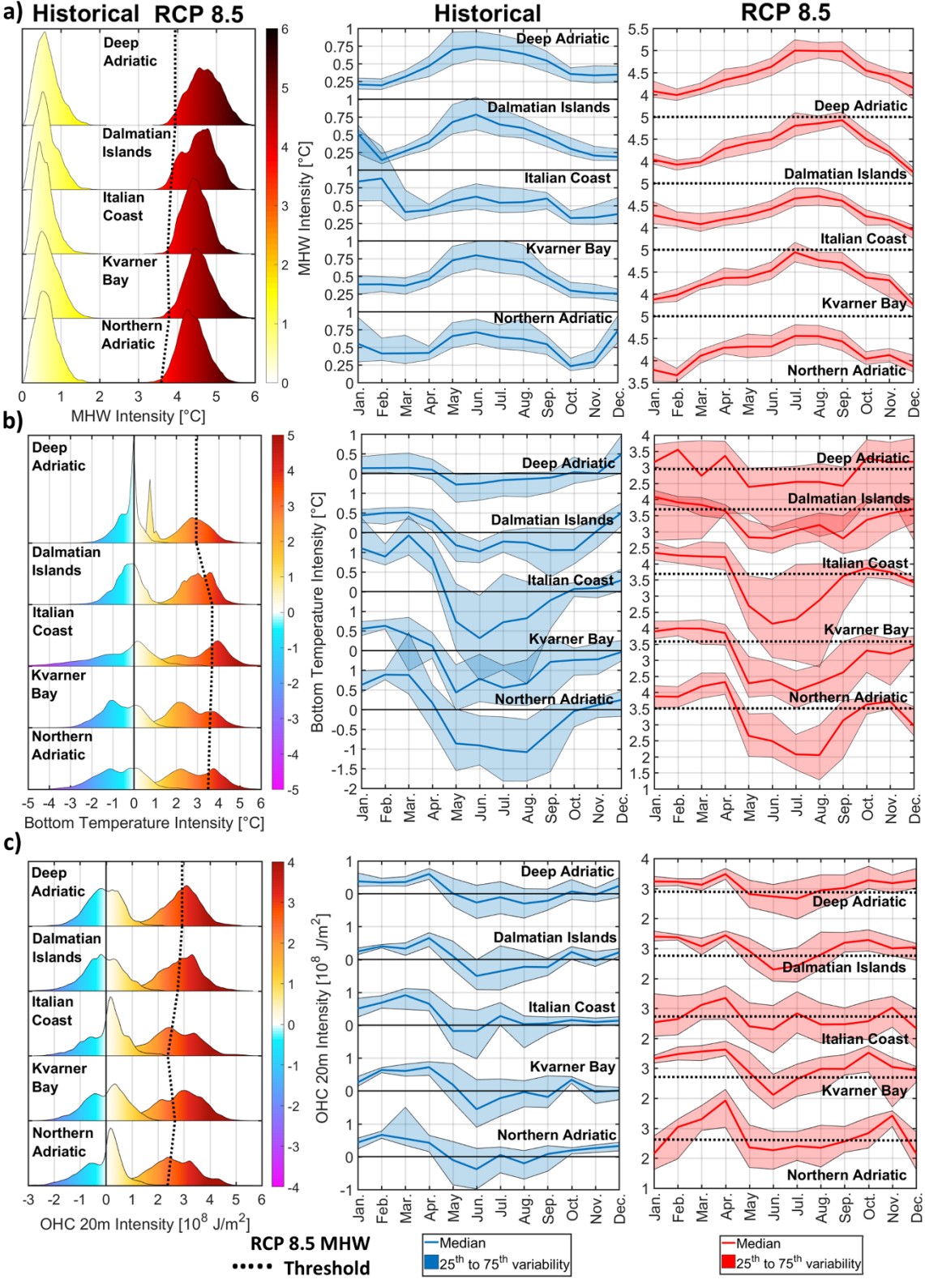

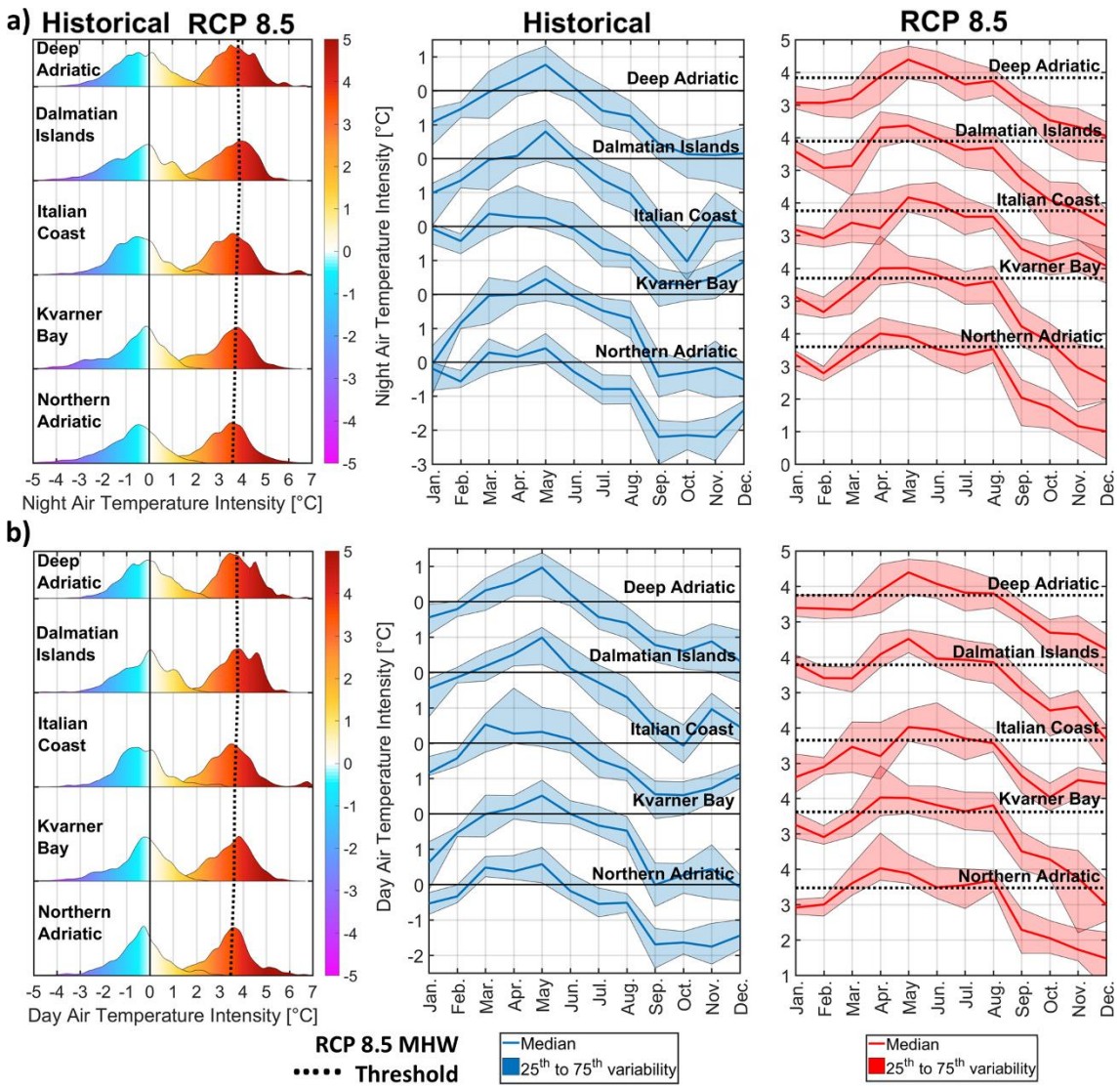

**Figure 8. For the 5 selected subdomains (Northern Adriatic, Kvarner Bay, Italian Coast, Dalmatian Islands and Deep Adriatic),**
**annual distributions (left panels) and monthly climatologies of the median, 25th and 75th percentiles (centre and right panels) of the**
**night (a) and day (b) air temperature intensities defined over the 31 years for the historical and RCP 8.5 conditions. All probability**
**density distributions are normalized to unit area to facilitate direct comparison.**

temperature threshold is defined for all depths within the Deep Adriatic subdomain, which has an average depth of

approximately 300 m, no definitive conclusions about bottom MHWs can be drawn for the deepest part of the Adriatic Sea

(South Adriatic Pit; Fig 1). This also explains the 2 °C spread (between the 25th and 75th percentiles) of the bottom temperatures seen in the monthly climatology within the Deep Adriatic subdomain.

**Table 1. Proportion (in percent) of occurrences of historical and RCP 8.5 bottom ocean temperature, OHC at 20 m as well as night and day air temperatures above the historical and RCP 8.5 thresholds, respectively.**

| | Bottom ocean temperatures | | OHC at 20 m | | Night air temperatures | | Day air temperatures | |
|---|---|---|---|---|---|---|---|---|
| | Historical | RCP 8.5 | Historical | RCP 8.5 | Historical | RCP 8.5 | Historical | RCP 8.5 |
| Northern Adriatic | 30 % | 32 % | 54 % | 51 % | 28 % | 40 % | 32 % | 48 % |
| Kvarner Bay | 28 % | 23 % | 54 % | 60 % | 33 % | 40 % | 37 % | 48 % |
| Italian Coast | 42 % | 43 % | 62 % | 57 % | 29 % | 37 % | 32 % | 43 % |
| Dalmatian Islands | 32 % | 20 % | 45 % | 65 % | 34 % | 39 % | 44 % | 52 % |
| Deep Adriatic | 32 % | 32 % | 47 % | 51 % | 37 % | 44 % | 43 % | 50 % |

In terms of the monthly climatologies, for all the subdomains, most of the MHW mean intensities vary between 0.25 °C and 1
°C under the historical conditions and between 3.5 °C and 5 °C for the RCP 8.5 scenario with the highest values reached between April and September. However, for the Italian Coast and Northern Adriatic subdomains, the highest MHW intensities are also reached in December and January, particularly under the historical conditions. The monthly climatologies of both bottom temperature and OHC at 20 m present similar variations for all the subdomains. Under the historical conditions, they exceed their threshold—i.e., their extreme coincide with surface MHWs—mostly in winter between November and April.
Under the RCP 8.5 conditions, in the Northern Adriatic and Italian Coast subdomains, bottom temperatures and OHC at 20 m experience the impact of surface MHWs between February and May as well as between September and November but tend to be below their thresholds in December and/or January. Further, the Italian Coast subdomain presents the largest variations in bottom temperature intensities (more than 2.5 °C) during summer between May and September. However, in the Kvarner Bay, Dalmatian Islands and Deep Adriatic subdomains, the OHC at 20 m thresholds are surpassed between July and May which
means that surface MHWs influenced the first 20 m of these domains most of year. Additionally, in the Kvarner Bay and Dalmatian Islands subdomains, the bottom temperatures only surpass their threshold between January and April. Under both historical and RCP 8.5 conditions. In contrast with the ocean results, for all the subdomains, the night and day air temperature intensities mostly exceed their thresholds between March and July for both historical and RCP 8.5 conditions. Consequently, atmospheric heatwaves are more likely to occur during MHWs in Spring and Summer.
In order to further explore the connection between MHWs and atmospheric conditions under both historical and RCP 8.5 conditions, the contribution of total air-sea heat fluxes ($dSST_{Q_{net}}$) to the change in sea-surface temperature anomaly ($dSST_A$) is examined during the onset and decline of the MHWs. The results are presented as distributions over the entire Adriatic Sea (Fig. 9a), percentage of air-sea driven events within the four coastal subdomains (Dalmatian Islands, Kvarner Bay, Northern Adriatic and Italian Coast; Fig. 9b) and distributions of the events depending on the depth and location of the
subdomains (Fig. 10). Importantly, the percentages of events primarily driven by air-sea heat fluxes is calculated under the assumption that the contribution of total air-sea heat fluxes to the change in sea-surface temperature anomaly is considered

significant when more than half of the warming/cooling can be attributed to air-sea heat fluxes—i.e., $dSST_{Q_{net}} / dSST_A > 0.5$

.

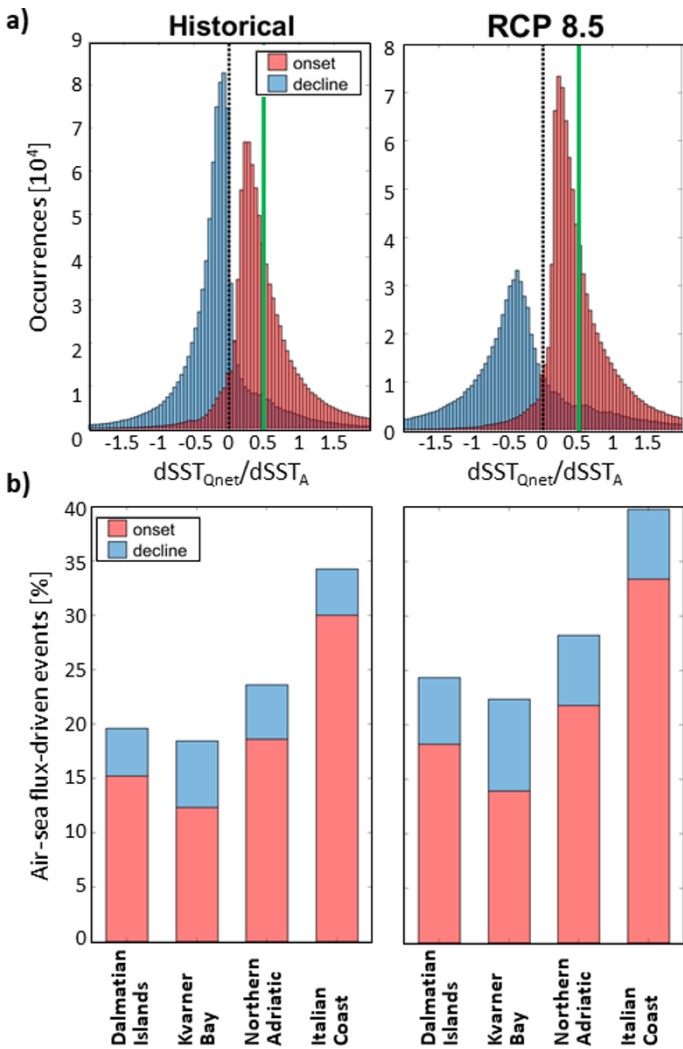

**Figure 9. Contribution of total air-sea heat fluxes ($dSST_{Q_{net}}$) to the change in sea-surface temperature anomaly ($dSST_A$) during MHW onset and decline phases along the nearshore areas of the Adriatic Sea (between 5 and 50 m depth) over the 31-year period of the historical and RCP 8.5 simulations (a). Percentage of events primarily driven by air-sea heat fluxes (i.e., with more than half of the warming/cooling attributed to air-sea heat fluxes) for the four nearshore subdomains (b).**

Under this assumption, MHWs are predominantly influenced by the air-sea fluxes during their onset phase compare to their decline for all subdomains and both scenarios. Furthermore, under both historical and RCP 8.5 scenarios, air-sea fluxes exert the strongest influence on MHWs within the Northern Adriatic (24 % and 28 %) and Italian Coast (34 % and 40 %) subdomains, while their impact is weaker in the Dalmatian Islands (20 % and 25 %) and Kvarner Bay (18 % and 23 %) subdomains.

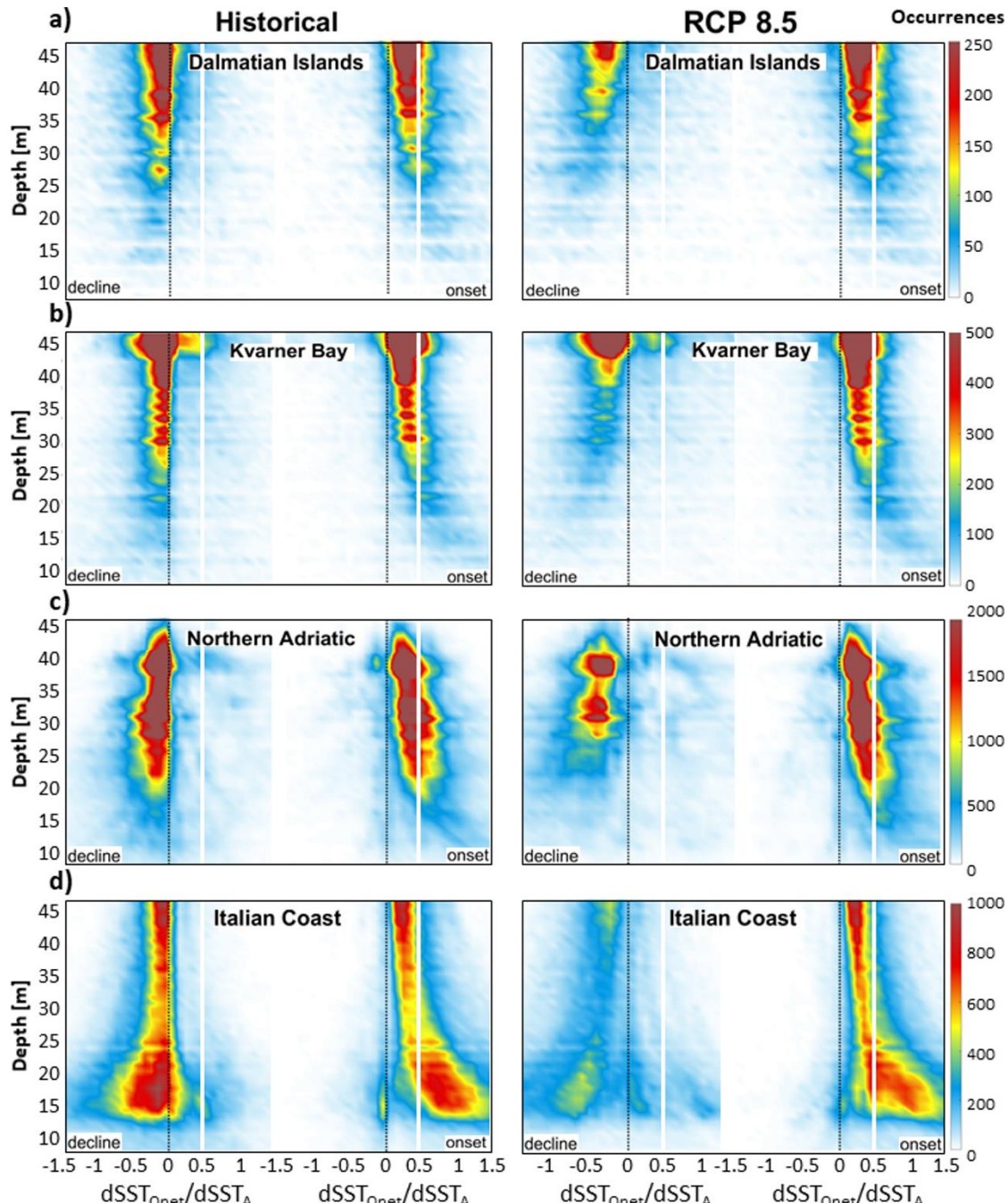

**Figure 10. Distribution of the contribution of total air-sea heat fluxes ($dSST_{Q_{net}}$) to the change in sea-surface temperature anomaly**
**($dSST_{A}$) during MHW onset and decline phases depending on the depths (between 5 and 50 m) of the four nearshore subdomains and over the 31-year long periods of the historical and RCP 8.5 simulations.**

Finally, as shown in Figure 10, air-sea fluxes tend to influence both the onset and decline of nearshore MHWs (i.e., at depths between 5 m and 20 m), particularly within the Northern Adriatic and Italian Coast subdomains. Under historical conditions, the distributions of air-sea influence during the onset and decline phases appear symmetrical, indicating that air-sea fluxes primarily impact the onset of MHWs. However, under RCP 8.5 conditions, this symmetry is lost, with an increasing number of events influenced by air-sea fluxes during the decline phase of MHWs.

## 4 Discussion

### 4.1 Performance of the AdriSC model

In terms of the capacity of the AdriSC model to reproduce historical MHW events, the strongest MHWs extracted above the 95th percentile of the AdriSC cumulative heat index results ($CHI \geq 3.7$ °C km² day; Fig 4e)—corresponding to years 1998, 2003, 2005, 2007, 2009, 2011, 2012, 2013, and 2015—can be confirmed by several observational studies. For instance, Sparnocchia et al. (2006) documented an extreme MHW during the summer of 2003 using in situ observations in the central Ligurian Sea, noting record-high sea-surface temperature anomalies that align with the AdriSC detection of 2003 as one of the longest and strongest Adriatic events. In a recent study, Martinez et al. (2023) analysed long-term sea-surface temperature (SST) records in the Mediterranean and identified significant MHW occurrences in 1998, 2003, 2007, 2012 and 2015, emphasizing the interannual variability of these events. It should be noted that Martinez et al. (2023) identified more Adriatic MHWs than the AdriSC model in the first half of the 1987-2017 period as they used detrended sea-surface temperature data. In the Adriatic, Salini et al. (2024) analysed the heatwaves along the Italian Adriatic coast between 2008 and 2022. They found that the 2009 MHW was the most intense event with more than 4 °C of anomaly peak intensity while the 2011 MHW was more intense in the northern Adriatic. Moreover, Garrabou et al. (2019) linked recurrent marine heatwaves across the Mediterranean to ecological impacts, noting substantial events during 2005, 2012, 2013, and 2015. This convergence of evidence suggests that the years flagged by the AdriSC model are consistent with observational studies.

Furthermore, the AdriSC results, showing that historical MHWs occurring in winter (from November to March) are more likely to affect the temperatures of the entire water column than summer events, are partially confirmed by the studies by Olita et al. (2007) and Garrabou et al. (2009). These studies demonstrated that the exceptional Mediterranean MHW in 2003 led to elevated sea temperatures during both summer and winter months that contributed to widespread mortality across various marine species including rocky benthic communities. Merryfield (2000) and Radko et al. (2014) have also demonstrated that, during summer, the Mediterranean Sea and, hence, the Adriatic Sea, experience strong thermal stratification, characterized by a warm, less dense surface layer overlying cooler, denser deep waters which inhibits vertical mixing, confining MHW effects to the surface layers. In winter, surface cooling and increased wind activity reduce stratification, promoting vertical mixing which allows MHWs to penetrate deeper into the water column, affecting subsurface and benthic ecosystems.

Concerning the projection of MHW events under extreme far-future warming, global studies project an increases in MHW intensity by up to approximately 50 % relative to historical values, while the duration of these events may double (Smale et al., 2019; Perkins-Kirkpatrick and Lewis, 2020). Further, the study of Konsta et al. (2025)—based on a global climate model with a horizontal resolution of about 11 km and using shifted baseline climatologies (similar to the presented study)—indicates that continued warming under RCP 8.5 could lead to shorter and milder shallow MHWs within the Mediterranean marine protected areas including the Adriatic Sea. In contrast, the AdriSC results based on the PGW methodology, suggest that under extreme far-future warming, Adriatic MHWs are likely to intensify and have a larger spatial extent while their duration remain identical to the historical conditions and their frequency slightly decrease. However, region-specific studies focusing solely on the Adriatic Sea remain limited.

Finally, during the summer season, under both historical and far-future extreme warming, the Deep Adriatic, Dalmatian Islands and Kvarner Bay subdomains are found to experience the strongest MHWs (Fig. 7a) which is confirmed by both observations and an ensemble of models in the study of Bonaldo et al. (2025). As highlighted by Glamuzina et al. (2024) the extreme warming already experienced in the Eastern Adriatic Sea leads to higher risk of invasiveness of non-native marine organisms. This increased invasiveness poses threats to native biodiversity and can disrupt existing fisheries and aquaculture operations. Additionally, extreme temperature events can cause mass mortality in marine species. For example, a significant die-off of Mediterranean mussels along the middle Adriatic coast already happened in summer 2022 and was linked to the effects of the climate change, highlighting the vulnerability of aquaculture operations to MHWs (Bracchetti et al., 2024).

## 4.2 Added value of the Pseudo-Global Warming method

In the preceding sections, MHWs were identified using an updated reference period for the RCP 8.5 scenario (Smith et al., 2025), which represents a standard approach when employing the Pseudo-Global Warming (PGW) methodology. In this subsection, MHWs are instead extracted using a detrended baseline following Smith et al. (2025), in order to better isolate and evaluate the influence of oceanic dynamics on the characteristics of MHWs.

### 4.2.1 MHW characteristics under the detrended baseline

The MHWs identified using the detrended baseline framework are first characterized by the number of events per year, along with the distributions of their mean intensity, duration, and spatial extent over the 31-year historical and RCP 8.5 simulations (Fig. 11a–d). Importantly, for consistency in comparison, the RCP 8.5 mean intensities are not adjusted to account for the difference between the RCP 8.5 and historical temperature climatologies.

In contrast to the previous results, the intensities of MHWs under historical and RCP 8.5 conditions are comparable, with median mean intensities of 0.59 °C and 0.63 °C, and 75th percentiles of 0.80 °C and 0.87 °C, respectively. Their spatial extent is also similar, with median areas of approximately 13,800 km² and 13,600 km², and 75th percentiles of about 20,400 km² and 21,400 km², respectively. However, as in the previous results, their duration remains consistent, with median values of 8 days and 75th percentiles of 12–13 days.

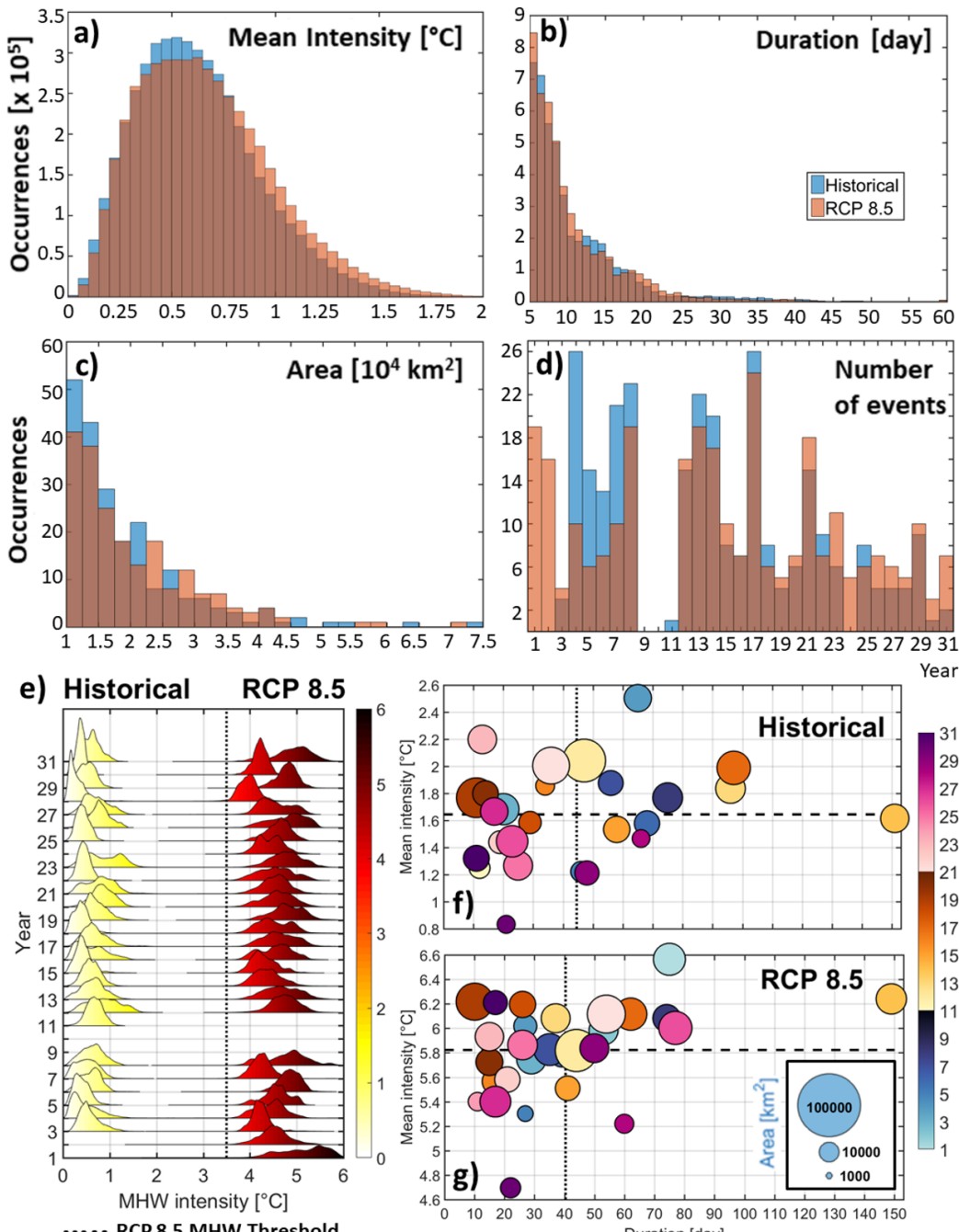

Figure 11. MHW characteristics based on the detrended sea-surface temperature. Distributions of the mean intensity as originally extracted with the heatwaveR package (a), duration (b) and area (c) as well as yearly time series of the number of events defined over the 31 years for the historical and RCP 8.5 conditions. Yearly probability density distributions of the mean intensity during the historical and RCP 8.5 31-year long periods (e). All probability density distributions are normalized to unit area to facilitate direct comparison. Scatter plots of the yearly maximum value of the mean intensity versus the maximum yearly value of the duration with the size of the circles depending on the maximum yearly value of the area during the historical (f) and RCP 8.5 (g) 31-year long periods.

Additionally, in contrast with the previous results, within the detrended baseline framework, MHWs are slightly more frequent under RCP 8.5 conditions, with a total of 287 events compared to 280 events under historical conditions. Notably, during the first eight years of the historical and RCP 8.5 simulations, the number of MHW events increases significantly under this framework—reaching 117 and 91 events, respectively—compared to just 26 and 38 events, respectively, in the previous (non-detrended) results. These results clearly highlight the influence of long-term temperature trends on the detection and characterization of marine heatwaves. In particular, the significant increase in the number of events during the early years of both simulations under the detrended baseline framework—compared to the original results—demonstrates that background warming trends in the Adriatic Sea can obscure or delay the identification of MHWs. By removing these trends, it becomes possible to isolate the intrinsic variability and short-term anomalies that define MHW events, thus offering a more consistent basis for defining the impact of the ocean dynamics on the MHWs.

Further characterization of MHWs under historical and RCP 8.5 conditions is carried out using yearly probability density distributions of MHW mean intensity (Fig. 11e) and scatter plots displaying the yearly maximum values of MHW mean intensity, duration, and spatial extent (Fig. 11f, g). The density plots reveal that, under historical conditions, no MHWs are detected in the first two years of the simulation—compared to the first three years in the previous (non-detrended) results. As in the previous analysis, no MHWs are identified in years 9–10 in either simulation; however, in year 11, one event is now detected under historical conditions (Fig. 11d). Under both historical and far-future extreme warming scenarios, the yearly maximum MHW durations (on average 42 and 40 days, respectively), maximum mean intensities (on average 1.6 °C and 5.6 °C, respectively), and maximum affected areas (mostly below 50,000 km²) remain comparable to those obtained with the non-detrended baseline. The most notable difference between the two approaches lies in the increased detection of more intense and longer MHWs during the first decade of both simulations when using the detrended baseline, highlighting again the impact of baseline trends on the timing and characteristics of MHW detection.

Overall, the comparison between detrended and non-detrended baselines underscores how baseline selection influences the interpretation of marine heatwave characteristics. While the non-detrended approach captures the compounded effects of long-term warming and short-term variability, the detrended baseline isolates interannual to decadal variability more effectively. This distinction provides complementary insights: the former emphasizes the trajectory of change under climate forcing, while the latter enhances the detection of underlying oceanic dynamics. Together, these methodologies offer a more comprehensive understanding of future MHW behaviour in a rapidly warming Adriatic context.

### 4.2.2 Impact of the Po River plume

For both historical and RCP 8.5 scenarios, the results presented in this study clearly highlight that the Northern Adriatic and Italian Coast subdomains exhibit seasonal variations in bottom temperature and ocean heat content (OHC) at 20 m that differ significantly from the rest of the Adriatic Sea. These regions also experience the strongest influence of air-sea fluxes on sea-surface temperatures, particularly during the onset phase of marine heatwaves (MHWs).

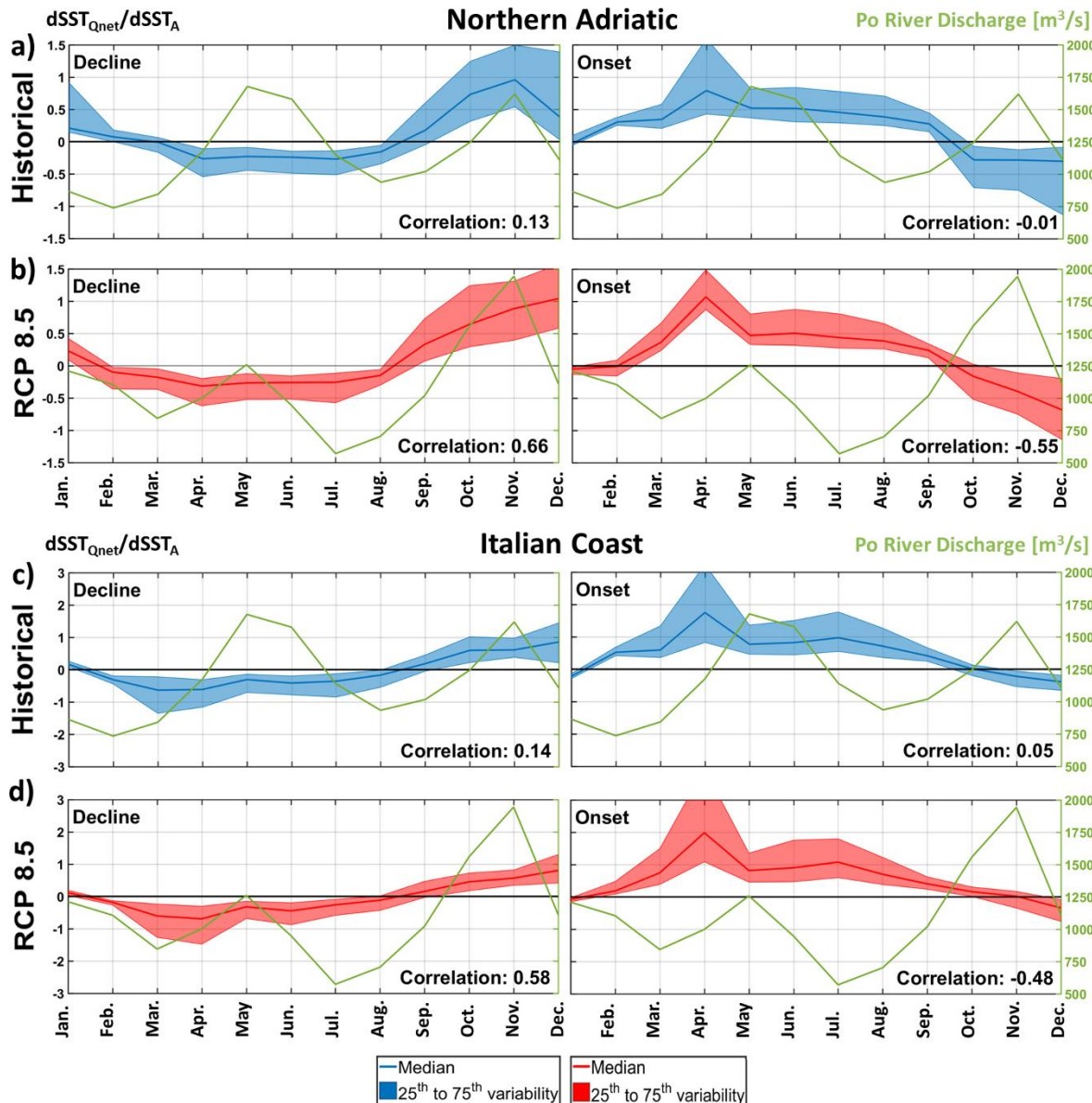

**Figure 12. For the Northern Adriatic (a, b) and Italian Coast (c, d) subdomains, monthly climatologies of the median, 25th and 75th percentiles of the contribution of total air-sea heat fluxes ($dSST_{Q_{net}}$) to the change in sea-surface temperature anomaly ($dSST_A$) during MHW onset and decline phases with the monthly climatology of the Po River discharge superimposed (in green) and defined over the 31 years for the historical (b, d, in blue) and RCP 8.5 (a, c, in red) conditions.**

In these areas, freshwater discharge from the Po River generates a buoyant plume that induces stratification, which, as noted by Verri et al. (2024), can act as a barrier to vertical mixing. This process potentially traps heat in the surface layers, contributing to elevated sea-surface temperatures. Conversely, during periods of reduced river discharge, weaker stratification may enhance vertical mixing, redistributing heat more evenly throughout the water column and potentially mitigating surface

warming. Additionally, Pranić et al. (2021) highlight that the Po River plume carries suspended sediments and organic matter, altering the optical properties of Northern Adriatic waters. These changes can influence the absorption of solar radiation, thereby affecting sea-surface temperatures. Areas with higher turbidity due to the plume may experience reduced solar penetration, leading to cooler surface waters, while clearer waters may allow for deeper solar penetration, affecting the thermal structure and potentially influencing the development and intensity of MHWs. Furthermore, the Po River discharge exhibits seasonal variability, with peak flows typically occurring in late autumn and spring, and lower flows in winter and summer. Sani et al. (2024) demonstrate that this variability influences circulation patterns in the Northern Adriatic Sea, affecting the distribution of water masses and heat content. During periods of high discharge, the expanded plume can modify local currents and stratification, potentially impacting the onset and duration of MHWs.

In the AdriSC historical simulation, the Po River discharge is modelled using historical observations, which indicate the lowest discharges (below 1,000 m³/s) in January–March and August–September, and the highest (above 1,500 m³/s) in May–June and November. Under the pseudo-global warming (PGW) approach used in the AdriSC RCP 8.5 simulation, historical discharges are increased by approximately 15 % between September and December and up to 50 % in January and February, while decreasing by up to 50 % between March and August—see Pranić et al. (2021) for more details. As a result, under RCP 8.5 conditions, the lowest discharges (below 1,000 m³/s) occur in March–April and June–September, while the highest discharges (above 1,500 m³/s) occur in October–November. Additionally, the temperature of the Po River plume is based on ground air temperature extracted from the ERA-Interim reanalysis near the delta, modified using the PGW approach for the RCP 8.5 scenario. The optical properties of the waters along the Po River plume remain identical under historical and RCP 8.5 conditions and have been corrected to better represent historical conditions.

After recalculation of the MHWs with the detrended sea-surface temperature signal, the AdriSC model results (Fig. 12) do not align with the conclusions of Verri et al. (2024). First, under historical conditions, no or weak correlations are found in the northern Adriatic (0.11) and along the Italian coast (0.05) between monthly MHW intensities and Po River discharge. However, between August and April, anti-correlations reach -0.54 in the northern Adriatic and -0.71 along the Italian coast. Additionally, under far-future extreme warming, the anti-correlations reach -0.49 in the northern Adriatic and -0.51 along the Italian coast. Second, as shown in Figure 12, for both the Northern Adriatic and Italian Coast subdomains, no correlation is found between the Po River discharge and $dSST_{Q_{net}}$ / $dSST_A$ during the MHW onset under historical conditions, while anti-correlations of approximately -0.5 are observed under the RCP 8.5 scenario. However, under historical conditions, between August and April, the anti-correlations reach -0.45 in the northern Adriatic and -0.21 along the Italian coast. For the decline of MHWs, positive correlations are observed: 0.13 and 0.66 in the northern Adriatic and 0.14 and 0.58 along the Italian coast for historical and RCP 8.5 conditions, respectively. However, between August and April, these correlations increase to 0.71 in the northern Adriatic and 0.56 along the Italian coast under historical conditions.

In summary, lower Po River discharge is more likely to coincide with the onset of intense MHWs, whereas higher discharges are associated with their decline. This relationship is mostly driven by air-sea fluxes between August and April under historical

conditions but persists year-round under far-future extreme warming. Furthermore, these MHWs extend to the bottom of the Adriatic Sea along the Po River plume between August and April under historical conditions, as well as from February to May and September to November under the RCP 8.5 scenario (Fig. 7, Northern Adriatic and Italian Coast subdomains).

These findings using the PGW approach suggest that MHWs are more likely to develop and persist under low Po River discharge conditions, when water clarity increases and solar radiation absorption is enhanced due to reduced suspended sediments and organic matter. Additionally, the temperature of the Po River plume may influence MHW occurrence, particularly during summer, when (anti-)correlations are weak, especially under historical conditions. In particular, the trends and variance of sea-surface temperatures calculated over the Adriatic Sea for the 31-year historical and RCP 8.5 periods (Fig. 3) reveal the influence of the Po River. Specifically, under both historical and RCP 8.5 conditions, a distinct pattern emerges along the path of the Po River plume in the northern Adriatic. Compared to other shallow areas of the basin, this region exhibits a slower rate of warming (0.45 °C/decade vs. 0.7 °C/decade under historical conditions, and 0.2 °C/decade vs. 0.35 °C/decade under RCP 8.5), coupled with markedly higher variance in sea-surface temperature (up to 40 °C² vs. 20 °C²). These results highlight the significant role of riverine influence in modulating local climate signals. Therefore, accurately representing Po River dynamics—including its temperature, suspended sediments, and organic matter—is crucial for predicting the impacts of climate change on MHW frequency and intensity in the Adriatic Sea.

### 4.2.3 Impact of significant oceanographic events

One of the most interesting features of the MHW results presented in this study is that, for both historical and far-future extreme warming conditions, no MHW is detected during years 9 to 11. After recalculating the MHWs using the detrended sea-surface temperature signal, this result is largely confirmed, except for the presence of a very weak MHW in year 11 under historical conditions (Fig. 11d, e and Fig. 13; CHI variability).  In the historical simulation, this period corresponds to the Eastern Mediterranean Transient (EMT) event which is characterized by massive dense water formation triggered by extreme heat losses and high salinity in the Aegean Sea during winter 1992-1993 (Roether et al., 1996, 2007; Klein et al., 1999; Velaoras et al., 2017). During the EMT, the northern Ionian Sea is filled with very dense water from the Aegean Sea, and the intrusion of Adriatic-originated water into the Levantine basin is blocked (Akpinar et al., 2016; Li and Tanhua, 2020). As explained in Denamiel et al. (2025), the RCP 8.5 scenario presented in this study is also forced with the historical EMT signal modified with an extreme warming climatological change. Consequently, the absence of MHW during this period could be attributed to the exceptional EMT conditions which have likely lowered the sea-surface temperatures and increased the vertical mixing in the Adriatic Sea and, hence, prevented extreme warming episodes under historical and RCP 8.5 conditions. Further, as the EMT is known to alter the thermohaline circulation, it may have strengthened cold-water upwelling or changed stratification patterns, leading to reduced heat retention at the surface. This disruption would have also limited the formation and persistence of MHWs in the Adriatic Sea during that period.

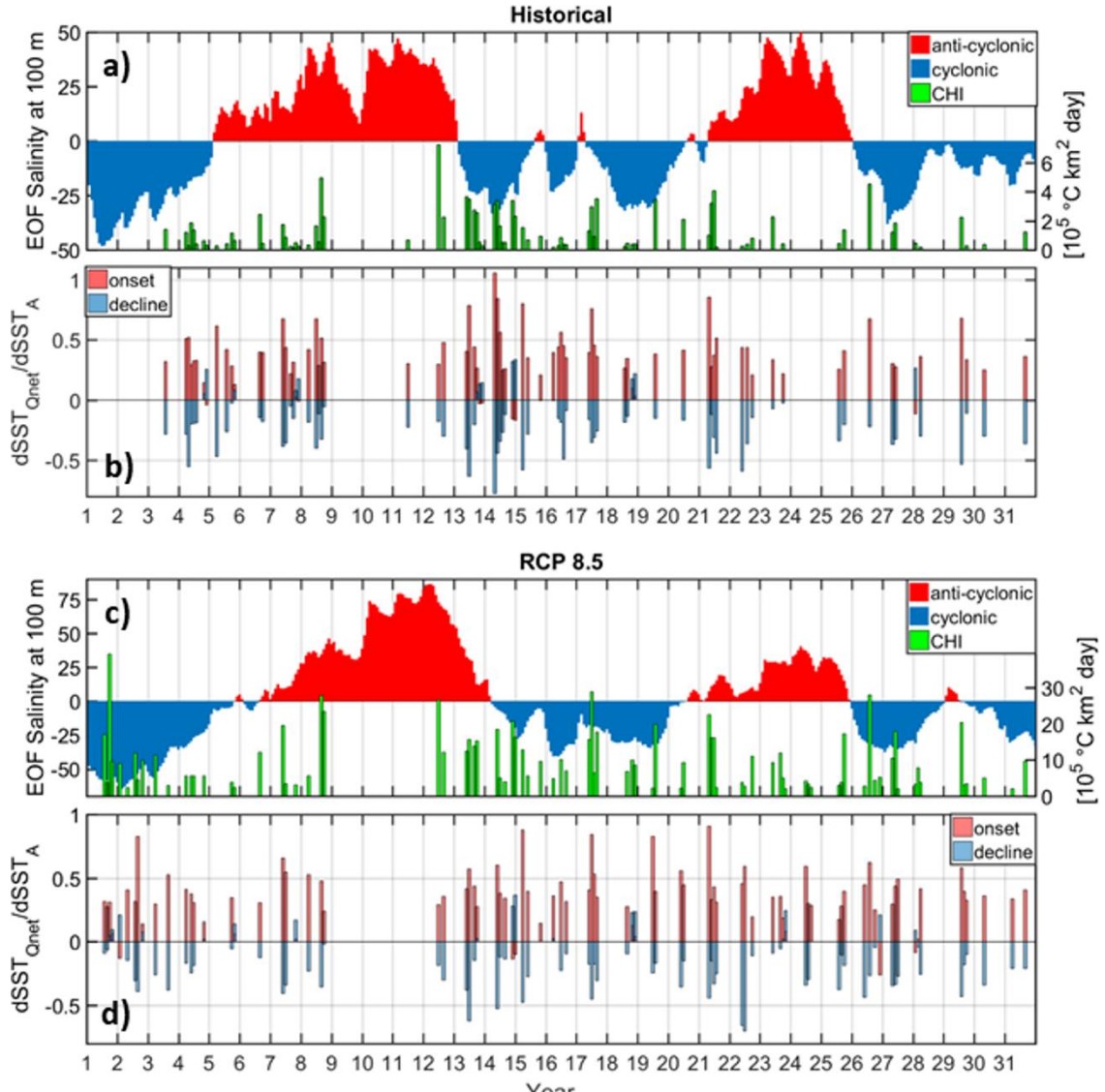

**Figure 13. Monthly time series over the 31-year Historical (a, b) and RCP 8.5 (c, d) periods, CHI—extracted from the detrended Adriatic SST—superimposed to the first Empirical Orthogonal Function (EOF) of salinity at 100 m depth representing the influence of the BiOS in the Adriatic Sea (a,c). Contribution of total air-sea heat fluxes ($dSST_{Q_{net}}$) to the change in sea-surface temperature anomaly ($dSST_A$) during MHW onset and decline phases.**

The presence of a gap in MHW activity visible under both historical and RCP 8.5 conditions, thus suggests that natural variability, such as the EMT, plays a significant role in modulating heatwave occurrences in the Adriatic, even under strong anthropogenic forcing. However, the study of Soto-Navarro et al. (2020) found that most Med-CORDEX regional climate models project a reduction of the dense water formation in the Aegean Sea and, hence, of the EMT-like situations. Finally, the suppression of MHW in the Adriatic Sea has not been previously linked to the EMT.

As EMT events lead to episodic and significant shifts in water properties, whereas BiOS represents a more gradual oscillatory pattern, these results suggest that while some large-scale circulation changes (like EMT) might influence MHW occurrence, others (like BiOS) may not exert a strong enough effect to suppress or enhance MHWs directly. Consequently, further targeted research is necessary to establish a definitive connection between MHW intensities and frequencies and EMT or other significant oceanographic events like the BiOS.

## 5 Conclusions

This study provides a comprehensive analysis of marine heatwaves (MHWs) in the Adriatic Sea under both historical (1987–2017) and far-future extreme warming (2070–2100, RCP 8.5) scenarios, utilizing the high-resolution Adriatic Sea and Coast (AdriSC) climate model. By employing the Pseudo-Global Warming (PGW) approach, the research offers valuable insights into the projected changes in MHW characteristics, their environmental drivers, and the implications for marine ecosystems and coastal communities. The findings highlight the increasing intensity and spatial extent of MHWs under extreme warming, while also revealing the limitations and strengths of the PGW methodology in simulating these events. The five key findings of the study are as follows.

First, under the RCP 8.5 scenario, MHWs in the Adriatic Sea are projected to become significantly more intense, with mean cumulative intensities increasing by approximately six times compared to historical conditions. The spatial extent of these events also expands, particularly in the central and eastern Adriatic regions. The Cumulative Heat Index (CHI) reveals that far-future MHWs will consistently exceed historical extremes, with values peaking at nearly $48 \times 10^5$ °C km² day, compared to $9 \times 10^5$ °C km² day in the historical period. This indicates a substantial rise in the severity of MHWs, posing greater risks to marine ecosystems and human activities.

Second, MHWs under both historical and far-future conditions exhibit strong seasonal patterns, with the most intense events occurring between May and September. However, under RCP 8.5, extreme MHWs are also projected to extend into late autumn and early winter, particularly in the northern Adriatic and along the Italian coast. The spatial distribution of MHWs remains consistent, with the deepest parts of the Adriatic experiencing the highest number of events. However, the nearshore areas, which are critical for aquaculture and tourism, are also significantly affected, albeit with fewer events.

Third, air-sea heat fluxes play an important role in the onset of MHWs. This influence is particularly strong in the northern Adriatic and along the Italian coast, where freshwater discharge from the Po River affects stratification, water quality and heat distribution. Under extreme warming, the influence of air-sea fluxes during the decline phase of MHWs increases, indicating a shift in the dynamics of heat dissipation. However, future MHWs are expected to have the same duration than under the historical conditions despite this altered heat exchange processes.

Fourth, the Po River plume significantly influences MHW dynamics, particularly in the northern Adriatic and along the Italian coast. However, the results presented in this study are not aligned with the previously published literature as the AdriSC simulations account for the changes in optical properties along the Po River plume which carries a lot of suspended sediments

and organic matter. Consequently, it is found that lower river discharge is associated with the onset of intense MHWs, as reduced stratification and increased solar penetration enhance surface warming. Conversely, higher discharge during the decline phase helps dissipate heat, mitigating the persistence of MHWs. Under RCP 8.5, the anti-correlation between Po River discharge and MHW intensity strengthens, highlighting the river's role in modulating heatwave dynamics under extreme warming.

Finally, for both simulations, the study identifies, for the very first time, a gap in MHW activity during the Eastern Mediterranean Transient (EMT). This event seems to suppress MHWs by enhancing vertical mixing and lowering sea-surface temperatures. However, while these results seem to demonstrates the influence of natural variability on MHWs, no correlation was found with the Ionian-Adriatic Bimodal Oscillating System (BiOS), highlighting the need for further research to clarify the role of oceanographic events in Adriatic MHW dynamics.

Based on these results, the added value of the Pseudo-Global Warming (PGW) approach—which effectively captures the thermodynamic changes associated with extreme warming—has been clearly demonstrated. However, PGW assumes stationarity in climate signals, potentially oversimplifying the evolving dynamics of MHWs. Furthermore, the one-way coupling between the atmosphere and ocean in the AdriSC model restricts the representation of feedback mechanisms which could influence MHW intensity and frequency. Future improvements in model coupling are thus necessary to enhance the accuracy of Adriatic MHW projections.

In conclusion, this study advances our understanding of MHWs in the Adriatic Sea under one extreme warming scenario, highlighting the critical role of high-resolution modelling in predicting future climate impacts. While the PGW approach provides valuable insights, its limitations emphasize the need for continued refinement of climate models and methodologies. Additionally, to provide a more comprehensive understanding of the cumulative impacts on marine ecosystems and to better represent the climate uncertainties, MHW projections under various scenarios should be derived by using an ensemble of PGW forcing and combined with assessments of other climate stressors, such as ocean acidification and deoxygenation.

Overall, the projected increase in MHW intensity and spatial extent under the on-going and future climate warming will pose significant threats to Adriatic marine ecosystems, including habitat degradation, biodiversity loss, and shifts in species distribution. Coastal communities reliant on fisheries and aquaculture will face heightened risks, as MHWs can lead to mass mortality events and disrupt marine resource availability. The findings thus underscore the urgent need to engage local stakeholders, including fisheries, aquaculture operators, and coastal managers, in the development of adaptive strategies to mitigate these impacts, particularly in vulnerable nearshore areas.

**Code availability**

The code of the COAWST model as well as the ecFlow pre-processing scripts and the input data needed to re-run the AdriSC climate model can be obtained under the Open Science Framework (OSF) data repository (Denamiel, 2021) under the MIT license.

**Data availability**

The model results used to produce this article can be obtained under the Open Science Framework (OSF) FAIR data repository (Denamiel, 2025a) under the CC-By Attribution 4.0 International license.

**Video supplement**

Movie S1. Animation of the historical (1987-2017) Adriatic marine heatwave mean intensity, duration and category extracted from the AdriSC model (Denamiel, 2025b; under the CC-By Attribution 4.0 International license).
Movie S2. Animation of the RCP 8.5 (2070-2100) Adriatic marine heatwave mean intensity, duration and category extracted from the AdriSC model (Denamiel, 2025b; under the CC-By Attribution 4.0 International license).

**Author contribution**

For this study, CD was solely responsible for the conceptualization, data curation, funding acquisition, investigation, methodology, project administration, resources, software, formal analysis and visualization and writing – original draft preparation.

**Competing interests**

The author declares that she has no conflict of interest.

**Acknowledgments**

The author would like to sincerely thank the anonymous reviewer and Dr. Justino Martinez for their thorough and constructive reviews. Their insightful comments and suggestions greatly helped improve the clarity, structure, and scientific quality of this manuscript. The computing and archive facilities used in this research were provided by the European Centre for Medium-range Weather Forecasts (ECMWF) through Croatian national quota and the ECMWF Special Projects "The Adriatic decadal
and inter-annual oscillations: modelling component" and "Numerical modelling of the Adriatic-Ionian decadal and inter-annual oscillations: from realistic simulations to process-oriented experiments".

**Financial Support**

The research has been supported by the HORIZON EUROHPC JU project ChEESE-2P (Grant 101093038).

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
