# Peer review of "Far-Future Climate Projection of the Adriatic Marine Heatwaves: a kilometre-scale experiment under extreme warming"

_EGUsphere, 2025_

## Author Comment (AC1)

**Response to reviewer #1 Justino Martinez**

I am truly grateful for the contribution of Reviewer #1—Dr. Justino Martinez—to enhancing the quality of the article. The comments are addressed in detail below (responses marked with "**R:**").

- **General comments**

*The Mediterranean, and the Adriatic in particular, is severely affected by global warming. Over the last 40 years, its surface has warmed by more than 1.5ºC [1] and up to 2ºC in some regions of the Adriatic. The deep waters are also expected to warm to a greater or lesser extent. Then, even for shorter periods than 31 years, the trend in the temperature series cannot be neglected. And this fact introduces my main objection to the publication of this work as is. For time series with trend, the calculation of the extremes cannot be performed using the original time series. Such a calculation leads to an overestimation or underestimation of the extremes, depending on the characteristics of the trend and the position of the extremes in the time series. This is especially important if the value of extremes (their deviation from the mean or the median) are comparable to the accumulated trend in the series. The non-stationarity, including trends and seasonality, of time series can invalidate the assumptions of many extreme value models*

[…]

*In the light of all this evidence, I think that in order to study the phenomenon occurring in some years of the time series (such as the influence of the EMT in the MHW intensity), it is necessary to use the detrended time series (historical and RCP 8.5). This is not difficult as the Schlegel R package can be fed with the detrended time series instead of the original time series.*

*The MHW community is making efforts to establish a common definition of MHW, mainly based on the baseline used. The recent work by Smith et al [8] is a first step in this direction, defining different baselines (i.e. different MHW detection methods) according to the aim of the study (ecological risk of MHW, effect of variability changes...). In particular, the fixed baseline (the one used by the author in the work under revision) is the recommended for intercomparisons between periods (the case under study in this case) but not for variability changes derived from discharges of rivers (Po) or temporal intrusion of deep waters (EMT).*
* * *
**R:** Following the advice of the reviewer, and based on the recent study of Smith et al. (2025), the article will now use the detrended SST to extract Adriatic MHWs to assess the variability changes derived from ocean dynamics (Po River, EMT and BiOS).

First, the following introduction will be added to paragraph "4.2 Added value of the Pseudo-Global Warming method":

*"This section highlights the added value of the Pseudo-Global Warming (PGW) approach implemented in the AdriSC kilometre-scale model by evaluating the influence of Adriatic Sea dynamics on marine heatwave (MHW) variability. Following the methodologies of Amaya et al. (2023) and Smith et al. (2025), both MHWs and the contributions of the total air-sea heat fluxes ( $dSST_{Q_{net}}$ ) to the change in sea-surface temperature anomaly ( $dSST_A$ ) have been recalculated using detrended sea surface temperature signals."*

[Figure]

**Figure 9.** For the Northern Adriatic (a, b) and Italian Coast (c, d) subdomains, monthly climatologies of the median, 25th and 75th percentiles of the contribution of total air-sea heat fluxes ($dSST_{Q_{net}}$) to the change in sea-surface temperature anomaly ($dSST_A$) during MHW onset and decline phases with the monthly climatology of the Po River discharge superimposed (in green) and defined over the 31 years for the historical (b, d, in blue) and RCP 8.5 (a, c, in red) conditions.

Second, Figure 9 will be updated with the new results and the analysis of the results will be updated as follows:

*"After recalculation of the MHWs with the detrended sea-surface temperature signal, the AdriSC model results do not align with the conclusions of Verri et al. (2024). First, under historical conditions, no or weak correlations are found in the northern Adriatic (0.11) and along the Italian coast (0.05) between monthly MHW intensities and Po River discharge. However, between August and April, anti-correlations reach -0.54 in the northern Adriatic and -0.71 along the Italian coast.*

*Additionally, under far-future extreme warming, the anti-correlations reach -0.49 in the northern Adriatic and -0.51 along the Italian coast. Second, as shown in Figure 9, for both the Northern Adriatic and Italian Coast subdomains, no correlation is found between the Po River discharge and $dSST_{Q_{net}}$ / $dSST_A$ during the MHW onset under historical conditions, while anti-correlations of approximately -0.5 are observed under the RCP 8.5 scenario. However, under historical conditions, between August and April, the anti-correlations reach -0.45 in the northern Adriatic and -0.21 along the Italian coast. For the decline of MHWs, positive correlations are observed: 0.13 and 0.66 in the northern Adriatic and 0.14 and 0.58 along the Italian coast for historical and RCP 8.5 conditions, respectively. However, between August and April, these correlations increase to 0.71 in the northern Adriatic and 0.56 along the Italian coast under historical conditions."*

It should be noted that, despite some changes in the correlations, these results confirm the previous findings and the conclusions remain unchanged concerning the interactions between Po river plume and Adriatic MHWs—particularly the need to better represent the Po river dynamics including its temperature, suspended sediments, and organic matter.

Third, for the interactions of Adriatic MHWs with the major oceanographic events, the following paragraph and figure will be added:

*"One of the most interesting features of the MHW results presented in this study is that, for both historical and far-future extreme warming conditions, no MHW is detected during years 9 to 11. After recalculating the MHWs using the detrended sea surface temperature signal, this result is largely confirmed, except for the presence of a very weak MHW in year 11 under historical conditions (Fig. 10; CHI variability)."*

The following discussion concerning both EMT and BiOS based on the new figure 10 will also be added:

*"The presence of a gap in MHW activity visible under both historical and RCP 8.5 conditions, thus suggests that natural variability, such as the EMT, plays a significant role in modulating heatwave occurrences in the Adriatic, even under strong anthropogenic forcing. However, the study of Soto-Navarro et al. (2020) found that most Med-CORDEX regional climate models project a reduction of the dense water formation in the Aegean Sea and, hence, of the EMT-like situations. Finally, the suppression of MHW in the Adriatic Sea has not been previously linked to the EMT.*

*Additionally, the study by Parras-Berrocal et al. (2023) suggested that major oceanographic events, such as the EMT, which affect thermohaline circulation and dense water formation, can influence the occurrence and characteristics of MHWs in the Mediterranean Sea. Consequently, it is reasonable to assume that Adriatic MHW variability may also be affected by the Ionian-Adriatic Bimodal Oscillating System (BiOS; Gačić et al., 2010), which links the quasi-decadal reversals of the Northern Ionian Gyre circulation to salinity variability and dense water dynamics in the Adriatic Sea. Previous studies (Denamiel et al., 2022, 2025) have shown that, under both historical and RCP 8.5 conditions, the BiOS signal in the Ionian Sea is strongly correlated—at a 2-year lag—with the first empirical orthogonal function (EOF) of salinity at 100 m depth, which can therefore serve as a proxy for BiOS-induced variability in the Adriatic Sea (Fig. 10a, c).*

*However, despite a slight increase in both the frequency of MHWs and the influence of air-sea fluxes on their onset during the cyclonic phases of the BiOS under historical and RCP 8.5 scenarios, no definitive connection is found between the Cumulative Heat Index (CHI) and the BiOS-induced variability in the Adriatic Sea."*

[Figure]

**Figure 10. Monthly time series over the 31-year Historical (a, b) and RCP 8.5 (c, d) periods, CHI—extracted from the detrended Adriatic SST—superimposed to the first Empirical Orthogonal Function (EOF) of salinity at 100 m depth representing the influence of the BiOS in the Adriatic Sea (a,c). Contribution of total air-sea heat fluxes ( $dSST_{Q_{net}}$ ) to the change in sea-surface temperature anomaly ( $dSST_A$ ) during MHW onset and decline phases.**

As previously, the main conclusions remain unchanged.

- **Specific comments**

*The influence of the Po outflow on the presence, intensity or extent of the MHW is a very interesting (and expected) result. If you finally agree to calculate the temperature trend for historical periods and RCP8.5, you may be able to relate the Po outflow and the Western Adriatic Current to a slower rate of warming. It could be an interesting new paper if you can relate Po outflow to lower warming rates as a function of depth using a km-scale experiment. This is something we can see intuitively using a global Mediterranean Sea surface temperature product (see Figure 1 of the paper you referred to as Martinez et al 2023 in your manuscript). A large version of this figure can be found in*

https://www.frontiersin.org/files/Articles/1193164/fmars-10-1193164-HTML/image_m/fmars-10-1193164-g001.jpg
* * *
**R:** The reviewer's suggestion is deeply appreciated and opens a particularly stimulating perspective for future work. Based on the previous AdriSC model results published in Tojčić et al. (2023, 2024), we have already observed that SST variability along the Po River plume may exert an even more pronounced influence than the long-term SST trends themselves (see Fig. R1 below). Additionally, the high concentration of suspended matter transported by the Po River can significantly alters heat penetration into the water column, which is another factor worth exploring.

[Figure]

Figure R1. Trend and variance of the sea surface temperature (SST) during the 31-year of the historical and RCP 8.5 period.

My current view is that a robust understanding of the Po plume's impact on Adriatic MHWs would require an accurate and high-resolution representation of plume dynamics (including suspended matter). That said, I greatly value the reviewer's insight and would be very happy to explore this further, should the reviewer be interested in collaborating on a dedicated study focusing on these interactions.

*Following our paper (Martinez et al 2023), note that you have identified the significant MHW of the second half of the period, but not the earlier ones, most likely due to the fixed baseline you have used.*

**R:** The following paragraph will be added to address this comment:

*"It should be noted that Martinez et al. (2023) identified more Adriatic MHWs than the AdriSC model in the first half of the 1987-2017 period as they used detrended sea-surface temperature data."*

- **Technical comments**

*• Figure 3e: The probability density functions are normalized to have unity area? Same question for figures 5 and 6*

**R:** This is correct and the methodology section will be modified as follows:

*"All annual and monthly probability density functions of the historical and RCP 8.5 intensities are normalized to have a unity area following a kernel-smoothing method (Bowman and Azzalini, 1997) evaluated for 100 equally spaced points."*

*• Lline 260: You mention a double peak in temperature in the Deep Adriatic subdomain. Are you referring to figure 5b?*

**R:** This is correct and the sentence will be modified as follows:

*"Notably, the presence of a double peak in the distribution of ocean bottom temperature intensities in the Deep Adriatic subdomain (at about 0.5 °C and 3 °C; Fig. 5b) is linked to the fact that, under the RCP 8.5 scenario, the shallow areas of the Adriatic Sea are expected to warm by up to 3.5 °C, while the deepest parts only by up to 0.5 °C (Denamiel et al., 2025)."*

**References:**

Amaya, D.J., Jacox, M.G., Fewings, M.R., Saba, V.S., Stuecker, M.F., Rykaczewski, R.R., Ross, A.C., Stock, C.A., Capotondi, A., Petrik, C.M., Bograd, S.J., Alexander, M.A., Cheng, W., Hermann, A.J., Kearney, K.A., and Powell, B.S.: Marine heatwaves need clear definitions so coastal communities can adapt, Nature, 616, 29-32, https://doi.org/10.1038/d41586-023-00924-2, 2023.

Gačić, M., Borzelli, G.E., Civitarese, G., Cardin, V., and Yari, S.: Can internal processes sustain reversals of the ocean upper circulation? The Ionian Sea example, Geophysical Research Letters, 37(9), https://doi.org/10.1029/2010GL043216, 2010.

Martínez, J., Leonelli, F.E., García-Ladona, E., Garrabou, J., Kersting, D.K., Bensoussan, N., and Pisano, A.: Evolution of marine heatwaves in warming seas: the Mediterranean Sea case study, Front. Mar. Sci., 10, 1193164, https://doi.org/10.3389/fmars.2023.1193164, 2023.

Parras-Berrocal, I.M., Vázquez, R., Cabos, W., Sein, D.V., Álvarez, O., Bruno, M., and Izquierdo, A.: Dense water formation in the eastern Mediterranean under a global warming scenario, Ocean Science, 19, 941–952, https://doi.org/10.5194/os-19-941-2023, 2023.

Smith, K.E., Sen Gupta, A., Amaya, D., Benthuysen, J.A., Burrows, M.T., Capotondi, A., Filbee-Dexter, K., Frölicher, T.L., Hobday, A.J., Holbrook, N.J., Malan, N., Moore, P.J., Oliver, E.C.J., Richaud, B., Salcedo-Castro, J., Smale, D.A., Thomsen, M., and Wernberg, T.: Baseline matters: Challenges and implications of different marine heatwave baselines, Progress in Oceanography, 231, 103404, https://doi.org/10.1016/j.pocean.2024.103404, 2025.

Tojčić, I., Denamiel, C., and Vilibić, I.: Kilometer-scale trends and variability of the Adriatic present climate (1987–2017), Climate Dynamics, 61, 2521–2545, https://doi.org/10.1007/s00382-023-06700-2, 2023.

Tojčić, I., Denamiel, C., and Vilibić, I.: Kilometer-scale trends, variability, and extremes of the Adriatic far-future climate (RCP 8.5, 2070−2100), Frontiers in Marine Science, 11, 1329020. https://doi.org/fmars.2024.1329020, 2024.

Verri, G., Furnari, L., Gunduz, M., Senatore, A., Santos da Costa, V., De Lorenzis, A., Fedele, G., Manco, I., Mentaschi, L., Clementi, E., Coppini, G., Mercogliano, P., Mendicino, G. and Pinardi, N.: Climate projections of the Adriatic Sea: role of river release, Front. Clim., 6, 1368413, https://doi.org/10.3389/fclim.2024.1368413, 2024.

---

## Author Comment (AC2)

**Response to reviewer #1 Justino Martinez**

Responses are marked with "**R:**".
* * *
*1.- A brief explanation about how the trend has been computed*

**R:** The following paragraph will be added to section 4.2:

*"In this study, trends are calculated using the Theil-Sen estimation method (Mondal et al., 2012), which is robust to outliers and often more accurate than simple linear regression when applied to skewed or heteroskedastic data. This method also performs competitively with non-robust least squares regression in terms of statistical power, even for normally distributed datasets. Additionally, the non-parametric Mann-Kendall test (originally proposed by Mann, 1945; further developed by Kendall, 1975; and Gilbert, 1987) is employed to detect the presence of monotonic (linear or non-linear) trends by evaluating whether a time series shows a consistent increase, decrease, or no change. In the case of sea-surface temperatures in the Adriatic Sea, all trends are found to be statistically significant under both historical and RCP 8.5 conditions (Tojčić et al., 2023, 2024)."*

*2.- Figures 2, 3, and 4 should be modified because a detrended is now performed. Please let me know if this is not true.*

**R:** Based on the reviewer's previous suggestion — *"Then, keeping the current baseline to perform the intercomparison between historical and RCP8.5 periods, and including the detrended baseline to study possible influence of Po outflow or EMT is, in my opinion, the correct approach to the study"* — my interpretation was that Figures 2 to 8 could remain unchanged, as they specifically aim to compare MHWs between the historical and RCP 8.5 scenarios using a consistent baseline. Consequently, I have only updated the figures in Section 4.2, which focus on the influence of ocean dynamics and required the use of the detrended SST signal.

However, please do not hesitate to let me know if I have misunderstood your intent, and I will be happy to revise the figures accordingly.
* * *
*3.- The figure R1 (trend) should be incorporated to the paper. Additionally, the trend could be shown only with the positive values of the bar (range 0:0.5) to show clearly the zones more affected by warming*

**R:** The following paragraph and figure will be added to section 4.2.1:

*"In particular, the trends and variance of sea surface temperatures calculated over the Adriatic Sea for the 31-year historical and RCP 8.5 periods (Fig. 10) reveal the influence of the Po River. Specifically, under both historical and RCP 8.5 conditions, a distinct pattern emerges along the path of the Po River plume in the northern Adriatic. Compared to other shallow areas of the basin, this region exhibits a slower rate of warming (0.45 °C/decade vs. 0.7 °C/decade under historical*

*conditions, and 0.2 °C/decade vs. 0.35 °C/decade under RCP 8.5), coupled with markedly higher variance in sea-surface temperature (up to 40 °C² vs. 20 °C²). These results highlight the significant role of riverine influence in modulating local climate signals."*

[Figure]

**Figure 10. Trend (a, c) and variance (b, d) for the sea surface temperature over the Adriatic basin derived for the historical (a, b) and RCP 8.5 (c, d) 31-year long periods.**

**References:**

Gilbert, R.O.: Statistical methods for environmental pollution monitoring, Wiley, New York, 1987.

Kendall, M.G.: Rank correlation methods, 4th edn. Charles Griffin, London, 1975.

Mann, H.B.: Non-parametric tests against trend, Econometrica1, 3, 163–171, 1945.

---

## Author Comment (AC3)

**Response to reviewer #2**

I am truly grateful for the contribution of Reviewer #2 to enhancing the quality of the article. The comments are addressed in detail below (responses marked with "**R:**").

- **General comments**

*First, is that the author should better present, from the beginning, the methodology used (the Pseudo-Global Warming approach) together with the limitations of this approach. The description in the Methods section is rather succinct, and the presentation of what the limitations of this approach could be comes rather late in my opinion (it is mentioned in the introduction but the implications are not clear).*

**R:** I appreciate the reviewer's suggestion regarding the presentation of the Pseudo-Global Warming (PGW) methodology and its limitations. The initial intent was to discuss the limitations of the PGW approach after presenting the MHW results, so as to directly link the findings with the underlying assumptions and constraints of the methodology. However, following the reviewer's recommendation, I will include a dedicated subsection (i.e., 2.2 Pseudo-Global Warming (PGW) methodology) in the Methodology section that clearly presents the PGW approach and outlines its main limitations. This restructuring will allow for a clearer and earlier understanding of the methodological framework, while maintaining the connection between the results and their implications through targeted references in the results section.

*"2.2 Pseudo-Global Warming (PGW) methodology*

*As described and illustrated in Denamiel et al. (2025) and Figure 1, for the RCP 8.5 simulation, the Pseudo-Global Warming (PGW) approach (Schär et al., 1996; Denamiel et al., 2020a) is used to adjust the historical forcing dataset by incorporating climatological changes from the LMDZ4-NEMOMED8 RCSM (Hourdin et al., 2006; Beuvier et al., 2010). Specifically, atmospheric variables such as air temperature, relative humidity, and wind components from ERA-Interim are modified using differences between the 2070–2100 and 1987–2017 periods under RCP 8.5 (ΔT, ΔRH, ΔU, ΔV, respectively). These changes generate 6-hourly three-dimensional atmospheric forcing for all 366 days of the year, which are then applied to the WRF 15-km model in the PGW simulation. Similarly, oceanic variables, including temperature, salinity, and currents, are adjusted using the climatological differences to produce daily three-dimensional oceanic forcing for ROMS 3-km (ΔT ocean, ΔS ocean, ΔU ocean, ΔV ocean, respectively). This process ensures that each simulated year inherits the synoptic conditions of the historical reanalysis while embedding the projected climatological shifts.*

*However, the pseudo-global warming (PGW) approach has several limitations, including simplified atmosphere-ocean dynamics, the assumption of stationarity, perturbations in initial and boundary conditions, resolution constraints, and neglected feedback mechanisms. The PGW method used in the AdriSC climate model to predict marine heatwaves (MHWs) in the Adriatic Sea incorporates both thermodynamic changes—such as temperature, salinity, and humidity—and dynamic adjustments (i.e., wind and ocean currents). Consequently, the*

*inaccuracies in representing circulation patterns critical for heatwave development, as highlighted by Xue et al. (2023), are somewhat mitigated. However, the PGW approach assumes that the relationship between large-scale climate drivers and regional weather patterns remains constant over time (i.e., the same climatological changes are applied every year; Brogli et al., 2023). This assumption can lead to potential misrepresentations of future MHW characteristics. As Xue et al. (2023) point out, the effectiveness of the PGW method also depends on how perturbations (i.e., climatological changes) are applied to initial and boundary conditions. Inconsistent or inappropriate perturbations can result in significant variations in simulated outcomes, affecting the reliability of heatwave projections. Additionally, Heim et al. (2023) note that PGW is constrained by the resolution of the driving data and the capabilities of the regional model, which can impact the accurate representation of localized heatwave events. In this study, the initial and boundary conditions are derived from a coupled atmosphere-ocean Med-CORDEX regional climate model with a resolution of approximately 15 km, which is likely sufficient to capture the main dynamical properties of the Mediterranean Sea. However, the very short spin-up period (two months) for the AdriSC RCP 8.5 simulation is likely to influence the results, particularly in the first two to three years of the simulation.*

*Finally, by focusing on imposed large-scale changes, the PGW approach may overlook regional feedback processes, such as land-atmosphere interactions, which can influence heatwave intensity and frequency (Heim et al., 2023). A key limitation of the AdriSC climate model is that it employs a one-way coupling between the atmosphere and the ocean—i.e., the sea surface temperature from the ocean model is not fed back into the atmospheric model."*
* * *
*My second comment is that there should be also a description of the differences between the historical and RCP8.5 runs, in terms of extrema, variability, intensity and trends. Without this we cannot put in context the differences observed between both runs presented in the paper.*
* * *
**R:** Thank you for this constructive suggestion. The differences between the historical and RCP 8.5 simulations in terms of sea surface temperature extrema, variability, intensity, and trends have been thoroughly analyzed in previous studies (Tojčić et al., 2023, 2024). However, I understand the importance of summarizing these key aspects directly within the current manuscript to provide a more complete context for the comparison of marine heatwave characteristics.

To address this, a dedicated paragraph in Section 2 will be added to summarize the main findings from Tojčić et al. (2023, 2024), and a new figure (Fig. 3) will be included to illustrate the trends and variance of sea surface temperature across the Adriatic Sea for both the historical and RCP 8.5 simulations, as follows:

*"For the AdriSC historical and RCP 8.5 simulations, Tojčić et al. (2023, 2024) conducted a comprehensive analysis of sea surface temperature trends, variance, and extremes, showing that all identified trends are statistically significant. Notably, as illustrated in Figure 3, the rate of surface ocean warming is 40 % higher during the historical period than under the far-future RCP 8.5 scenario. Nonetheless, Tojčić et al. (2024) demonstrated that in the far-future period, the Adriatic Sea is projected to experience at least 20 additional days of extreme heat per month compared to historical conditions. This increase is primarily driven by surface temperature anomalies (i.e., differences between RCP 8.5 and historical conditions) exceeding 3 °C, especially*

*in coastal regions. Regarding sea surface temperature variance (Fig. 3b, d), an average increase of 15 % is observed across the entire Adriatic Sea, with localized increases of up to 25 % along the south-eastern coastal regions. These changes further highlight the relevance of the methodology adopted to extract and analyse MHWs. For the comparison between historical and RCP 8.5 conditions, the consistent climatological baseline enables the accurate identification of MHWs, while the additive adjustment applied to the RCP 8.5 scenario helps isolate the heatwave signal from background warming. For the analysis of the influence of ocean dynamics on MHWs, the removal of the large sea surface temperature trends is necessary to properly characterise the link between dynamical features and extreme events."*

The following figure (now Figure 3) will also be added:

[Figure]

**Figure 3. Trend (a, c) and variance (b, d) for the sea-surface temperature over the Adriatic basin derived for the historical (a, b) and RCP 8.5 (c, d) 31-year long periods.**

• **Specific comments**

*Figure 1. I'd limit the bathymetry colorbar to the depths found in the Adriatic. As it is now, we can clearly see Thyrrhenian bathymetry which is irrelevant, and the Adriatic is rather homogeneous*

**R:** Figure 1 will be updated as follows:

[Figure]

**Figure 1. Spatial coverage and horizontal resolution of the different grids used in the AdriSC climate model Setup including the topo-bathymetry of the AdriSC 1-km model with the locations of the 5 subdomains (coloured polygons) used in this study and the Pseudo-Global Warming temperature ocean forcing imposed in the AdriSC 3-km model southern boundary for the extreme warming simulation.**

*Line 125-129. I find it confusing what it is mentioned about the shifting of the RCP8.5 results: is this only for visualisation purposes? Why is this done?*

**R:** Thank you for your comment and the opportunity to clarify. The shifting of RCP 8.5 Marine Heatwave (MHW) intensities by the difference between RCP 8.5 and historical daily mean climatologies is not merely for visualization purposes, but rather a methodological choice commonly used in MHW studies to enable meaningful comparisons across climatological baselines (e.g., Deser et al., 2024).

Without this adjustment, MHW intensities under RCP 8.5 would reflect both the change in baseline temperature and the actual heatwave anomaly, making it difficult to disentangle the effects of background warming from the characteristics of the heatwaves themselves. By applying the shift, the MHW signal relative to its future mean state is isolated, which allows a more direct comparison with historical MHWs.

This approach will be better explained in the manuscript (Section 2) and illustrated with a new figure 2.

[Figure]

**Figure 2.** Illustration of the methodology used to extract the daily intensities of the sea-surface temperature (a), sea-bottom temperature (b), Ocean Heat Content (OHC) at 20 m (c), day air temperature (d) and night air temperature (e) during the Marine Heat Waves (MHW) extracted from both the Historical and RCP 8.5 runs.

The following explanation will be added to the text:

*"However, to simplify the comparison with the historical results, all the RCP 8.5 intensities presented in this article (except if mentioned explicitly) are shifted by $\Delta T_{c\lim}$, the difference between the RCP 8.5 and historical daily mean climatologies (hereafter RCP 8.5 MHW Threshold, as illustrated in Figure 2). This methodological adjustment is widely adopted in marine heatwave (MHW) research to enable meaningful intercomparisons across different climatological baselines (e.g., Deser et al., 2024). The rationale behind this approach is to separate the influence of background warming from the intrinsic properties of MHWs. Without such an adjustment, the intensities under RCP 8.5 would reflect both the elevated baseline temperatures and the heatwave*

*anomalies, making it difficult to isolate the climate change signal. By applying a shift equal to the difference between the historical and RCP 8.5 mean climatologies, the analysis focuses on the MHWs relative to their respective mean states, thereby allowing a clearer and more direct comparison of their characteristics across time periods. Additionally, although the RCP 8.5 MHW threshold exhibits pronounced seasonal variability, as shown in Figure 2, it remains effectively constant when considering annual distributions and can therefore be approximated by its 31-year average for year-round analyses."*

*Line 147: "All the intensities": which intensities*

**R:** To enhance the clarity, the paragraph will be rewritten as follows:

*"All the other intensities (as illustrated in Figure 2) are calculated by removing the daily historical 90th percentile for the ocean bottom temperatures as well as the OHCs and the daily historical 95th percentile (i.e., threshold generally used to detect daily atmospheric heatwaves; European State of the Climate, 2023) for the air temperatures during night and day from these results and averaging them over the duration the MHW."*

*Line 165: "deepest part": I'd suggest to add some bathymetry contours to Fig 2*

**R:** *"deepest part"* will be replaced by *"The Deep Adriatic subdomain (Fig. 1)"* as the area where the highest number of MHW events occur does in fact cover this subdomain.

*Line 168: central eastern Adriatic: maybe refer to the pre-defined regions?*

**R:** Excellent point. The text will be modified as follows:

*"with the highest and lowest values located within the Dalmatian Islands and Northern Adriatic subdomains, respectively."*

*Figure 3. Add (left) and (right) to the description of panel (e) in the caption. The colors used in panels (a) and (b) are rather random. I would suggest to use shades of the same colours for each decade (e.g. shades of blue for the first decade, shades of orange for second decade and shades of purple for the last one. I would avoid red and green on the same figure). What are the dotted lines in panel (e)? (one is straight and the other seems to follow the histograms). The colours of the histograms represent the same thing as the numbers on the x-axis? and what about the height of the histograms? Because I do not understand why the yellow histograms for the monthly panel are higher than the RCP8.5 histograms but are still yellow.*

**R:** The comments of the reviewer are addressed point by point below with the new figures:

Left and right will be added to the description while the colormap will be changed following the suggestion of the reviewer in both figures 2 & 3 (original draft) that will be figures 4 & 5 in the new version of the article in order to preserve the consistency on the way to present the results.

(continue to next pages)

[Figure]

**Figure 4. Spatial distributions of the total number of marine heatwave (MHW) events (a, b) and their associated mean cumulative intensity (c, d) across the Adriatic Sea for the historical period (a, c) and the RCP 8.5 scenario (b, d). Panels e and f show the Time series of Cumulative Heat Intensity (CHI) over the 31-year historical and RCP 8.5 periods (e, f).**

[Figure]

**Figure 5. Scatter plots of the yearly maximum value of the MHW mean intensity versus the maximum yearly value of the duration of the MHW with the size of the circles depending on the maximum yearly value of the area covered by the MHW during the historical (a) and RCP 8.5 (b) 31-year long periods. Monthly distributions of both the MHW categories---with each occurrence representing one cell of the model---presented as histograms and the number of MHW events during the historical (c) and RCP 8.5 (d) 31-year long periods. Yearly (left) and monthly (right) probability density distributions of the MHW mean intensity during the historical and RCP 8.5 31-year long periods (e).**

The dotted lines had a legend in the original figures but I admit that (1) the description of what they represent in the methodology was not properly detailed and (2) the legend might have been a bit small. The methodology section will be modified to address this deficiency (see text above which also explains why one line is straight and the other is curved) while the legend font will be increased in order for the information to be more visible.

Actually for the histograms, the colors represent the MHW categories as defined by Hobday et al. (2016) and explained in the methodology (with the traditional colors used to represent these categories). The x-axis represents the different months (as per the legend) and the y-axis represent the number of occurrences. This is a traditional way to represent a lot of information for the MHWs (e.g., Pastor and Khodayar, 2023). The fact that the historical histograms can be (as they are not for every month) higher than the RCP 8.5 histograms just reveals that more occurrences happen for this month while the categories show that the intensity of the MHWs is far greater in the RCP 8.5 scenario (brown and red colors corresponding to categories III & IV) than for the historical conditions (yellow and orange colors corresponding to categories I & II).

*Line 188. Here you mention for the first time that there are no MHWs for years 9-11 in both runs. This comes as a surprising fact, moreover since this happens in both runs, which makes it look suspicious that the PGW approach is not able to separate from the variability from the base/historical run. This is discussed in the Discussion section, but I think a brief comment (indicating this is fully described in the Discussion section) should be added here.*

**R:** The following paragraph will be added to the text: *"As explained in Denamiel et al. (2025), the large-scale atmospheric and oceanographic patterns that force the boundaries of the AdriSC WRF 15-km and ROMS 3-km grids under historical conditions are also present in the RCP 8.5 simulation. Consequently, the 3-year gap in MHW occurrence observed under both historical and RCP 8.5 conditions is likely linked to an extraordinary atmospheric or oceanographic event, which will be further discussed in Section 4."*

*Line 200. Is the word skewed the correct one here? For me skewed indicates there is a bias/error in the data, and not just that it show a trend towards a specific value (which is what is mentioned here).*

**R:** In statistics, "skewed" refers to a lack of symmetry in a probability distribution. A skewed distribution is asymmetrical, meaning the left and right sides are not mirror images of each other. It contrasts with a symmetrical distribution like a normal distribution, which has a perfectly symmetrical bell shape. Hence, in this context, the fact that the distributions are (right) skewed towards a certain value XX only means that the tail (the part extending beyond the peak) is on the right side of the distribution towards XX and that the mean is greater than the median.

To be more precise the word "right" will be added in front of skewed in the text.

*Line 222. Are MHWs less frequent in RCP8.5 because they are longer? (i.e. there is less time with no MHW so less possibilities for a new MHW to develop).*

**R:** This is an interesting comment. In fact, as suggested by reviewer #1, I have redone the analysis of the MHWs with the detrended sea surface temperature to better capture the impact of the ocean

dynamics on the MHW characteristics. Following this new analysis, more MHWs are picked up in the RCP 8.5 simulation than under historical conditions because the historical trends are 40 % higher than the RCP 8.5 trends (as shown in the new figure of trends and variance).

A new subsection and figure will be added in the discussion to present the detrended results and compare them with the previous (non-detrended results).

 *"**4.2 Added value of the Pseudo-Global Warming method***

[revised manuscript text omitted]

*Line 223-225. I do not understand what the author refers to in this sentence. If it is what I think I understand, doesn't a correlation of 0.9 show again that the PGW is not effectively separating events from the historical run from the RCP8.5?*

**R:** The reviewer is correct and this was originally analyzed in Section 4. As the limitation of the PGW will now be presented in Section 2, the paragraph will be modified as follows:

*"Furthermore, over the last 20 years of the simulations when MHWs occur annually, the correlations between median yearly variations in MHW intensity, area and duration under historical and RCP 8.5 conditions reach 0.92, 0.44 and 0.91, respectively. These high correlations between historical and RCP 8.5 MHW intensity, area, and duration are likely a consequence of the PGW approach–which assumes that the relationship between large-scale climate drivers and regional weather patterns remains constant over time–rather than realistic temporal features."*

*Figures 5 and 6. Again, there is a dotted line which is not described in the caption, and no units in the y-axis of panels a.b.c (left). What is bottom temperature and OHC \*intensity\* ?*

**R:** The new version of the methodology section as well the new Figure 2 will explain correctly (1) what the dotted are representing and (2) what bottom temperature and OHC 20m intensities are (see above modifications). As per the previous figure, the font of the legend of the dotted lines will be increased.

Regarding the absence of units in the distribution plots, I would like to respectfully point out that this convention is also followed in Figure 3 (which will become Figure 5 in the revised version). This choice aligns with common practices in the literature for presenting normalized probability density functions (PDFs), as illustrated, for example, in Pastor and Khodayar (2023). Since these PDFs are normalized to an area of one, including physical units on the vertical axis would not convey additional quantitative meaning.

The following sentence will be added to the legend of the Figures:

*"All probability density distributions are normalized to unit area to facilitate direct comparison."*

That said, if the reviewer strongly feels that including units would improve clarity, I would be happy to revise the labels for ex-Figures 5 and 6 accordingly. Unfortunately, for ex-Figure 3, the figure's design and formatting constraints make it difficult to incorporate units without compromising readability. I hope this explanation clarifies the reasoning behind the current presentation.

(updated figures continue to next pages)

[Figure]

**Figure 7. For the 5 selected subdomains (Northern Adriatic, Kvarner Bay, Italian Coast, Dalmatian Islands and Deep Adriatic), distributions (left panels) and monthly climatologies of the median, 25th and 75th percentiles (centre and right panels) of the MHW (a), ocean bottom temperature (b) and OHC at 20 m (c) intensities**

defined over the 31 years for the historical and RCP 8.5 conditions. All probability density distributions are normalized to unit area to facilitate direct comparison.

[Figure]

**Figure 8.** For the 5 selected subdomains (Northern Adriatic, Kvarner Bay, Italian Coast, Dalmatian Islands and Deep Adriatic), annual distributions (left panels) and monthly climatologies of the median, 25[th] and 75[th] percentiles (centre and right panels) of the night (a) and day (b) air temperature intensities defined over the 31 years for the historical and RCP 8.5 conditions. All probability density distributions are normalized to unit area to facilitate direct comparison.

*Line 270. values of 0.25degC etc "above the climatology" I guess?*

**R:** With the clarification of the methodology in Section 2 (including what will be Figure 2 in the new version of the article), the meaning of "intensity" throughout the article should now be totally transparent.

*Figure 7. There is no description in the text of how the "percentage of events primarily driven by air-sea fluxes" is determined.*

**R:** The following sentence will be added in order to explain how the percentages are calculated: *"Importantly, the percentages of events primarily driven by air-sea heat fluxes is calculated under the assumption that the contribution of total air-sea heat fluxes to the change in sea-surface temperature anomaly is considered significant when more than half of the warming/cooling can be attributed to air-sea heat fluxes—i.e., $dSST_{Q_{net}} / dSST_A > 0.5$."*

*Line 326. I would not call these other studies "independent observations" since they might use the same datasets. Just "other studies" would be more adequate.*

**R:** The word *"independent"* will be removed and only *"observational studies"* will be kept.

*Line 374 and following: how are the percentage of changes in the discharges decided? Why is this not discussed in the Methods section?*

**R:** Thank you for this suggestion. The percentage changes in river discharges are part of the PGW methodology implemented in the AdriSC model and are fully described in Pranić et al. (2021). While I understand the importance of providing clarity on key modeling choices, especially for readers who may not be familiar with the AdriSC modelling suite, I believe it is also essential to build upon previous work without reproducing the entire methodological framework in each new article.

That being said, I acknowledge that the reference to Pranić et al. (2021) was missing in the original description, and I will revise the paragraph to include the following clarification:

*"– see Pranić et al. (2021) for more details."*

I hope this addition addresses the concern and provides the necessary guidance to readers seeking further information.

*Section 4.2.2 Here the author mentions the Easter Mediterranean Transient as a possible cause of the absence of MHW in years 9-11, but no proof or demonstration is given, so this is just a suggestion provided by the author that needs to be verified. Line 427 says no previous work has linked EMT to MHW suppression, but this work doesn't provide the link either. And again, the fact that the MHW absence is also noticed in the RCP8.5 run should be better explained as it reflects in my opinion that the PGW fails to detach from the base variability. There is some mention in line 447 but this should be made clearer before.*

**R:** I fully agree and this is why the following sentence concludes the subsection:

*"Consequently,* **further targeted research is necessary to establish a definitive connection** *between MHW intensities and frequencies and EMT or other significant oceanographic events like the BiOS."*

And also in the conclusion:

*"However, while these results* **seem** *to demonstrates the influence of natural variability on MHWs, no correlation was found with the Ionian-Adriatic Bimodal Oscillating System (BiOS), highlighting the need for* **further research to clarify the role of oceanographic events in Adriatic MHW dynamics.***"*

Further, with the addition of the limitations of the PGW method in section 2 and the sentence concerning the MHW gap (see reply to *Line 188* comment), I think the PGW framework will now be better explained.

Finally, while I appreciate the reviewer's perspective, I respectfully disagree that the absence of MHWs in the RCP 8.5 simulation indicates a shortcoming of the PGW methodology. On the contrary, it is precisely due to the controlled framework of the PGW approach that this particular event could be robustly identified and analyzed. By preserving the large-scale oceanographic and atmospheric variability from the historical boundary conditions, the PGW methodology enables a meaningful attribution of such anomalies, which may otherwise be obscured in fully coupled future projections.

**Additional References:**

Deser, C., Phillips, A.S., Alexander, M.A., Amaya, D.J., Capotondi, A., Jacox, M.G., and Scott, J.D.: Future Changes in the Intensity and Duration of Marine Heat and Cold Waves: Insights from Coupled Model Initial-Condition Large Ensembles, J. Climate, 37, 1877–1902, https://doi.org/10.1175/JCLI-D-23-0278.1, 2024.

Pastor, F. and Khodayar, S.: Marine heat waves: Characterizing a major climate impact in the Mediterranean, Science of The Total Environment, 861, 160621, https://doi.org/10.1016/j.scitotenv.2022.160621, 2023.